# Customizing a self-healing soft pump for robot

Wei Tang [1], Chao Zhang [1✉], Yiding Zhong[1], Pingan Zhu[1], Yu Hu[1], Zhongdong Jiao [1], Xiaofeng Wei[1], Gang Lu[1], Jinrong Wang [1], Yuwen Liang[1], Yangqiao Lin [1], Wei Wang[1], Huayong Yang[1] & Jun Zou [1✉]

Recent advances in soft materials enable robots to possess safer human-machine interaction ways and adaptive motions, yet there remain substantial challenges to develop universal driving power sources that can achieve performance trade-offs between actuation, speed, portability, and reliability in untethered applications. Here, we introduce a class of fully soft electronic pumps that utilize electrical energy to pump liquid through electrons and ions migration mechanism. Soft pumps combine good portability with excellent actuation performances. We develop special functional liquids that merge unique properties of electrically actuation and self-healing function, providing a direction for self-healing fluid power systems. Appearances and pumpabilities of soft pumps could be customized to meet personalized needs of diverse robots. Combined with a homemade miniature high-voltage power converter, two different soft pumps are implanted into robotic fish and vehicle to achieve their untethered motions, illustrating broad potential of soft pumps as universal power sources in untethered soft robotics.

---

[1] State Key Laboratory of Fluid Power and Mechatronic Systems, Zhejiang University, Hangzhou, China. ✉email: chao.zhang@zju.edu.cn; junzou@zju.edu.cn

Inspired by biological systems, scientists and engineers are increasingly interested in developing soft robots[1–4] capable of reproducing biological functionalities such as flexible skin-like perception and muscle-like actuation, providing more adaptable and safer ways for human-machine interaction, industrial automation, and healthcare, etc., than traditional rigid robots. To actuate soft robots, various soft actuation techniques[5–7], including pressure[8–10], thermal[11,12], magnetism[13,14], light[15], combustion[16,17], phase transition[18], etc., have been exploited. Traditional pneumatic/hydraulic soft actuators[9] are the most prevalent, yet the requirement of external bulky compressors/pumps[19] leads to their poor portability, prohibiting their untethered applications. Soft combustion-driven pumps[17] can provide the high-speed response and large generated pressure, but these are difficult to control and reuse. Thermal-activated phase transition[18] driving power sources have the advantages of large output force, but their response speeds are slow due to poor controllability of thermal. Magnetic-responsive or light-responsive[13–15] soft actuation has unique advantages in driving micro/nano-scale robots, but complex or bulky external equipment is always necessary. Although these existing techniques have some own unique merits in some respects, no technique has yet been the principal and universal driving power sources[2] to drive untethered soft robotic systems with performance trade-offs existing between powerful actuation, good controllability, rapid response, high reliability, and excellent portability, etc.

In nature, spiders have their unique biological hydraulic systems[20,21], allowing them to achieve autonomous and agile motions by directly utilizing the hydraulic power of hemolymph flow. The hydraulic power of hemolymph flow is generated by the hearts' pumping function, and then the pressurized hemolymph flows in their foot to achieve their movements, as shown in Fig. 1a. Whereas other animals (such as dogs, cats, etc.) use their hearts to pump blood so as to transport nutrients, and their limbs' movement is driven by muscles, not directly driven by the pressurized blood from their hearts. Furthermore, natural evolution enables the blood/hemolymph of biological systems to autonomous self-heal of their damaged tissues or skins[22], greatly improving their adaptability to unpredictable environments and increasing their longevity. Is it possible for soft robots to possess driving power sources mimicking the functions of spiders' hearts that are fully soft, powerful, built-in, diverse, long-term, quiet, and even self-healing, driving their untethered motions and healing their damages?

Recent stretchable electrohydrodynamic pumps[23] that can be embedded in soft robots provide a good way to solve the poor portability problem of traditional soft fluidic actuators, which can mimic the pumping function of spiders' hearts. Nevertheless, their flow rate and output force should be further improved to meet the actuation requirements of most untethered soft robotic applications. To mimic both pumping and self-healing functions of spiders' hearts, here we introduce a class of fully soft electronic pumps that take advantage of electron and ion migration mechanism to pump liquid under applied electric field and are capable of automatically healing the damages of soft robots with help of self-healing liquid. These soft pumps show some distinctive features: (i) They are not only lightweight and portable but also exhibit powerful and controllable actuation capabilities; (ii) their appearances could be easily customized to desired ones and implanted into different untethered soft robotic systems; (iii) their self-healing liquids can repair the damaged regions of fluidic systems, greatly improving the reliability of systems. Moreover, a homemade lightweight and miniature high-voltage power converter powered by battery or wireless power transfer system are developed to actuate and control the reversible bidirectional pumping of soft electronic pumps, enabling a soft robotic fish and a robotic vehicle to achieve untethered and versatile motions when the customized soft pumps are implanted into them. Soft electronic pumps offer a platform for developing fully soft, powerful, robust, rapid response, long-term, built-in, and universal driving power sources to drive untethered soft robots.

## Results

**Architectures of soft electronic pumps**. Spider has its built-in bio-hydraulic system, which provides the basis for the realization of its biological functions. Spider's bio-hydraulic system[20,21] utilizes its tubular heart to pump hemolymph to its foot, thereby converting the hydraulic energy of hemolymph into the kinetic energy of its agile motions, as shown in Fig. 1a. Inspired by spider's heart, we introduce a class of fully soft, built-in electronic pumps to actuate untethered soft robots. Figure 1b shows a prototype of soft electronic pumps, consisting of a soft ring grounding electrode with four circular holes (Conductive silicone 60A), two soft ring positive electrodes with four needles (Conductive silicone 50A), two insulated electrode supports (Silicone 30A), and an insulated elastomer shell (Silicone 5A or polydimethylsiloxane (PDMS)). All-soft-matter polymeric architectures allow soft pumps to be stretchable (Supplementary Video 1) and fabricated by widely available soft materials (e.g., silicone, hydrogel, etc.) as well as basic fabrication techniques (e.g., traditional casting, multi-material 3D printing, etc.). Supplementary Fig. 1 shows the details of the fabrication process of soft pumps. Each soft needle in positive electrodes is aligned with the center of each hole in the grounding electrode, eventually forming the electrode pair modular unit of soft pumps—needle-hole electrode pair. Supplementary Fig. 2 shows three types of soft electronic pumps with different shells and electrode configurations, where the lightest pump weighs only 3 g.

When soft electronic pumps are filled with functional liquids, a strong jet flow could be generated from a needle tip through a hole under an applied electrical field. Note that the prototype of the soft electronic pump is made of four needle-hole modular units in parallel. Two positive electrodes are symmetrically distributed on either side of the grounding electrode, allowing for reversibly bidirectional pumping (Fig. 1c) when two opposite needle-hole electrode pairs are selectively applied with a voltage. Figure 1d, Supplementary Fig. 3, and Supplementary Video 2 demonstrate the rapid, controllable, and fast-switching bidirectional pumping processes of a soft electronic pump under different applied voltages.

**The operational principle of soft electronic pumps**. The operational principle of soft electronic pumps is that (Fig. 1e): Under a strong non-uniform electric field between positive and grounding electrodes, electrons in a small amount of neutral liquid molecules near needle positive electrodes will overcome potential barriers and separate from liquid molecules to become free electrons. These free electrons with negative charges will be absorbed into needle positive electrodes from the liquid. The original neutral liquid molecules become positive ions due to the loss of electrons, the so-called positively charged process. Under Coulomb forces[24,25], these free positive ions move along the electric field lines from needle positive electrodes to grounding electrodes, directly dragging neutral liquid molecules flow along with them through the hole to form a strong jet. As these free ions with positive charges reach to grounding electrodes, the electrons on grounding electrode surfaces will combine with these free ions to re-form neutral liquid molecules, the so-called negatively discharged process. Continuous liquid flow in soft electronic pumps will occur all the time due to the electron and ion migration under the applied electric field; while the flow

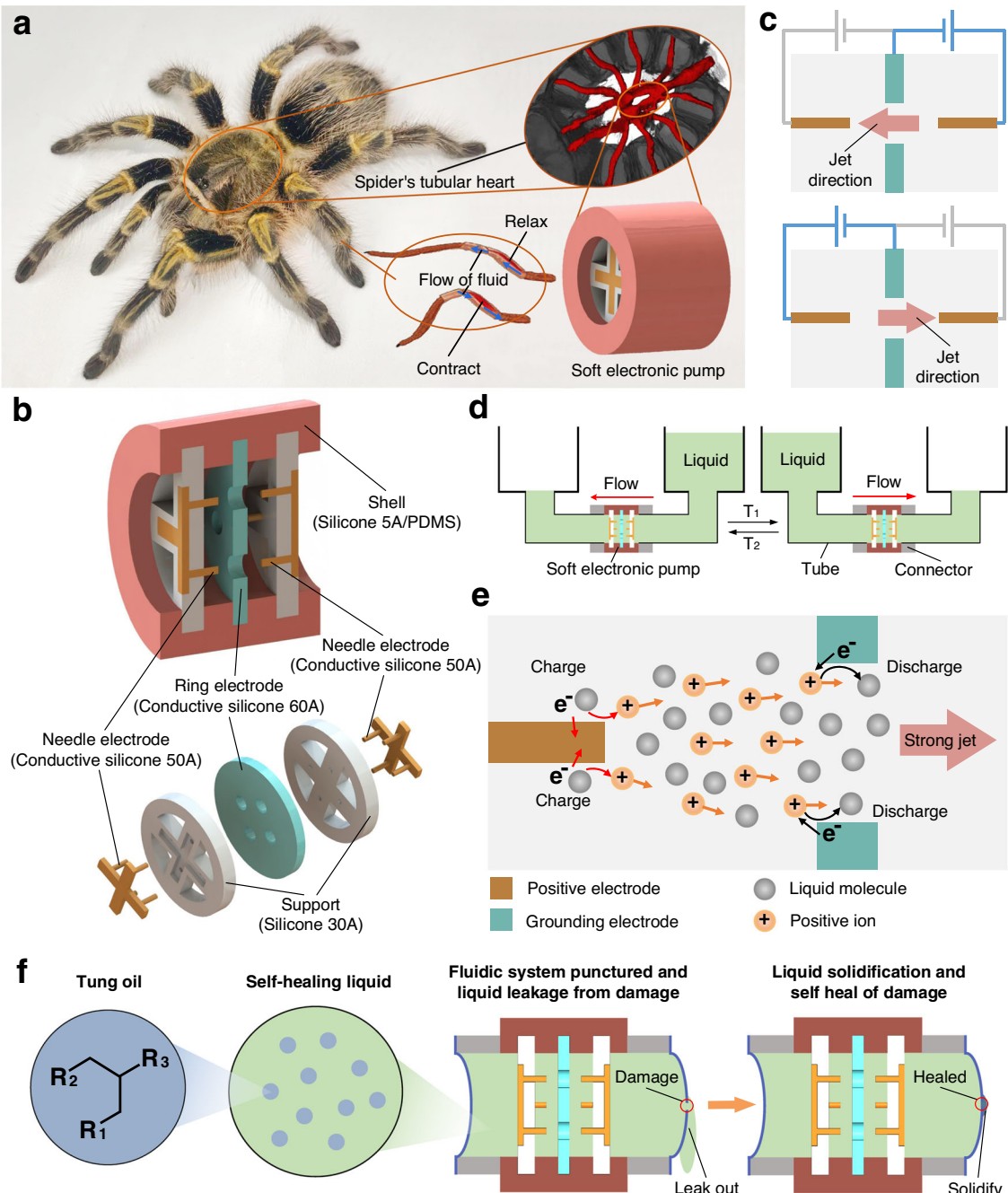

**Fig. 1 Bio-inspiration, architectures, operational principle, and self-healing liquid of soft electronic pumps. a** The built-in bio-hydraulic actuation system of the spider. Spider utilizes its tubular heart to pump the hemolymph to its foot, thereby driving the foot joint to extend. **b** Architectures of soft electronic pumps. All the components of pumps, including needle electrodes, ring electrodes, supports, and shells, are made of soft materials, which make the pump fully soft and stretchable. Needle-hole electrode pair in pumps forming the minimum electrode pair modular unit is used to move liquid. **c** Schematic of pump's reversibly bidirectional pumping capability. The pumping direction can be reversibly changed by selectively applying with voltages. **d** Schematic of pumping liquid between two cylindrical liquid reservoirs by the soft electronic pump. The pumping time $T_1$ and $T_2$ can be controlled by adjusting the applied voltages. **e** Operational principle of soft electronic pumps. Free ions are generated by the positively charged process when applied to a strong electric field, and then these ions subjected to the action of Coulombic forces will move along the electric field lines from the positive electrode to grounding electrodes, directly dragging neutral liquid molecules flow to form a strong jet. The negatively discharged process re-forms these ions into neutral liquid molecules when they reach the grounding electrodes. **f** Schematic of self-healing liquid pumped by soft electronic pump to heal the damages. The self-healing liquid is fabricated by dissolving evenly tung oil into dibutyl sebacate, which can form a solid film when exposed to air.

will stop as the electric field is removed. Theoretical model and numerical simulation of the operational principle can be seen in "Methods" section and Supplementary Fig. 4, where the distributions of electric potential, charge density, and velocity in the soft electronic pump are presented.

**Self-healing liquid pumped by soft electronic pumps**. Biological organisms possess self-healing blood/hemolymph capable of autonomous healing their damaged tissues or skins from external injuries[22]. Intrinsic softness enables soft robots to adapt to uncertain, dynamic environments, but they are also susceptible to

cuts, shears, and punctures due to sharp objects and edges. Taking inspiration from self-healing biological blood/hemolymph, we report a self-healing liquid that merges the unique properties of electrical response actuation and autonomous self-heal of damages in soft robots. This functional liquid is a dibutyl sebacate-tung oil solution, in which the tung oil is dissolved evenly in the dibutyl sebacate. Tung oil has excellent solidification properties because of its special compositions: oxygen-linked fatty carboxylate residue ($R_1$, $R_2$, and $R_3$ in Fig. 1f) and reactive conjugated carbon–carbon double bonds[26,27]. When tung oil is exposed to air, a solid film could be formed[28] due to that the reactive conjugated carbon–carbon double bonds readily combine with oxygen easily absorbed by oxygen linked fatty carboxylate residue ($R_1$, $R_2$, and $R_3$) to undergo free radically initiated homopolymerization[27]. When soft robots are damaged, the self-healing liquid inside robotic bodies will be exposed to air and then solidified to automatically heal the damages. As shown in Fig. 1f and Supplementary Video 3, the work of the soft pump will cause liquid to leak out continuously from the damage of the fluidic system once punctured, but the pump can operate well again without any leakage when self-healing of the damage is finished. The temperature has an obvious effect on the self-healing time, and the self-healing time of the punctured damage is ~6 h at 35 °C and ~1 day at 24 °C, respectively. The healed fluidic system can work continuously for 2 h without any leakage, demonstrating that this self-healing method is effective and reliable. Linalyl acetate-tung oil solution has a similar self-healing effect. The self-healing process can be achieved in air, but cannot be achieved in water due to lack of contact with air in water. Viscoelastic properties of the self-healed film are measured by dynamic thermomechanical analysis (DMA) tests, as shown in Supplementary Fig. 5. The self-healed film is transparent and elastic. The storage modulus of the self-healed film is ~14.3 kPa at 20 °C, ~13.2 kPa at 25 °C, and ~10.6 kPa at 30 °C, respectively. The loss modulus of the self-healed film is ~ −3.1 kPa at 20 °C, ~−2.5 kPa at 25 °C, and ~−2.1 kPa at 30 °C, respectively. To further characterize the adhesion property between the self-healed film and the silicone film, we conduct the repeated tensile tests of a silicone film with local self-healed film, as shown in Supplementary Fig. 6a. The silicone film is repeatedly stretched 200 times with 20% strain per time. After tests, it is found that the silicone film and the self-healed film are still firmly bonded together, illustrating good adhesion property between the two films. The tensile stress-strain curves of pristine and self-healed samples are close, as shown in Supplementary Fig. 6b, illustrating that the difference in the mechanical properties between them is small.

**Customizable appearances and pumpabilities of soft electronic pumps**. Appearances (size, shape) and pumpabilities (generated pressure, flow rate) of spiders' hearts are always characterized by biodiversity, enabling them to have their unique biological functions to adapt to their living environment. Interestingly, the appearances and pumpabilities of soft electronic pumps are also diverse, which could be customized to desired ones for various soft robotic applications due to the flexible layout of modular needle-hole electrode configuration. Note that fully soft polymeric architectures of pumps are convenient for multi-material 3D printing, allowing them to be easily manufactured into various sizes (from microscale to macroscale) and shapes (from simple geometry to complicated shape). Various sizes of soft pumps could be designed and manufactured by changing circumferential cross-sectional areas and axial electrode series lengths (Fig. 2a and Supplementary Fig. 7a); while various shapes (e.g., triangular, pentagonal, plane structure, complicated heart-like shape, etc.)

could be made by changing electrode layout, support shape, and shell shape (Fig. 2b and Supplementary Fig. 7b). Customizable appearances enable soft pumps to meet the diverse size and shape demands of various soft robotic systems.

In addition to customizable appearances, the pumpabilities of soft electronic pumps could also be customized by changing applied voltage, electrode configuration, liquid type, or electrode series/parallel integration use, etc. Supplementary Fig. 8 shows experimental setups for testing the performances of soft electronic pumps. As shown in Fig. 3a, the generated pressure increases as a quadratic curve against the applied voltage, and the liquid type have an obvious effect on the generated pressure. The change of electrode configurations (including needle diameter, hole diameter, and electrode gap) also affect the generated pressure. Electrical field strength could be enhanced by decreasing the needle diameter (Supplementary Fig. 9a), hole diameter (Supplementary Fig. 9b), or inter-electrode gap distance (Supplementary Fig. 9c), thereby improving the generated pressure of soft pumps. Different combinations of electrode configurations and liquid types can achieve different generated pressures (Fig. 3a), where a lightweight (3 g) and miniature (0.68-cm thickness) soft pump (Supplementary Fig. 2c) can generate a powerful pressure of ~9.2 kPa. Soft pumps with lighter weight and larger output pressure could be further obtained by selecting more powerful liquids (lower viscosity, generating more positive ions while maintaining low conductivity) and more elaborate electrode configurations (electrode pairs possessing smaller needle diameter, smaller hole diameter, and smaller inter-electrode gap distance; and electrode materials possessing stronger conductivity). Series integration of multiple needle-hole electrode pairs could also raise the generated pressure, as shown in Fig. 3c.

The flow rate of soft electronic pumps could also be customized by changing applied voltage, electrode configuration, liquid type, or electrode parallel integration use, as illustrated in Fig. 3b, d, and Supplementary Fig. 7d–e. The flow rate increases as the increase in applied voltage (Fig. 3b) and the decrease in needle diameter (Supplementary Fig. 9d) or inter-electrode gap distance (Supplementary Fig. 9e). Different combinations of electrode configurations and liquid types can also achieve different flow rates, where the maximum flow rate of the soft pump is ~521 ml min$^{-1}$ (Fig. 3b). Parallel integration of multiple needle-hole electrode pairs could raise the flow rate of soft pumps, as shown in Fig. 3d. The response time (peak time) of the soft pump is ~0.45 s, and the switching response time of bidirectional pumping is ~0.58 s, as shown in Fig. 3e. The dynamic bidirectional pumpabilities of soft pumps under a 1 Hz switching square wave are shown in Supplementary Fig. 9f. The frequency limits of this pump are tested as ~10 Hz. We test the lifetime of soft pumps at an applied voltage of 15 kV for 4 h (Fig. 3f), illustrating the durability and reliability of soft pumps. It is worth noting that the electrodes would be passivated and then stabilized when high voltage was applied a longer time, thereby causing the pressure to decrease and then stabilizing.

Parameters of electrode configurations (including needle diameter, hole diameter, and electrode gap) have obvious effects on the pumping performance (generated pressure and flow rate). Supplementary Fig. 9a–c shows the influence of the needle diameter, hole diameter, and electrode gap on the generated pressure. It is apparent that the generated pressure increases with the decreasing of needle diameter (Supplementary Fig. 9a), hole diameter (Supplementary Fig. 9b), and electrode gap (Supplementary Fig. 9c), respectively. Supplementary Fig. 9d, e show the influence of the needle diameter and electrode gap on the flow rate. The flow rate increases with the decreasing of needle diameter (Supplementary Fig. 9d) and electrode gap (Supplementary Fig. 9e), respectively.

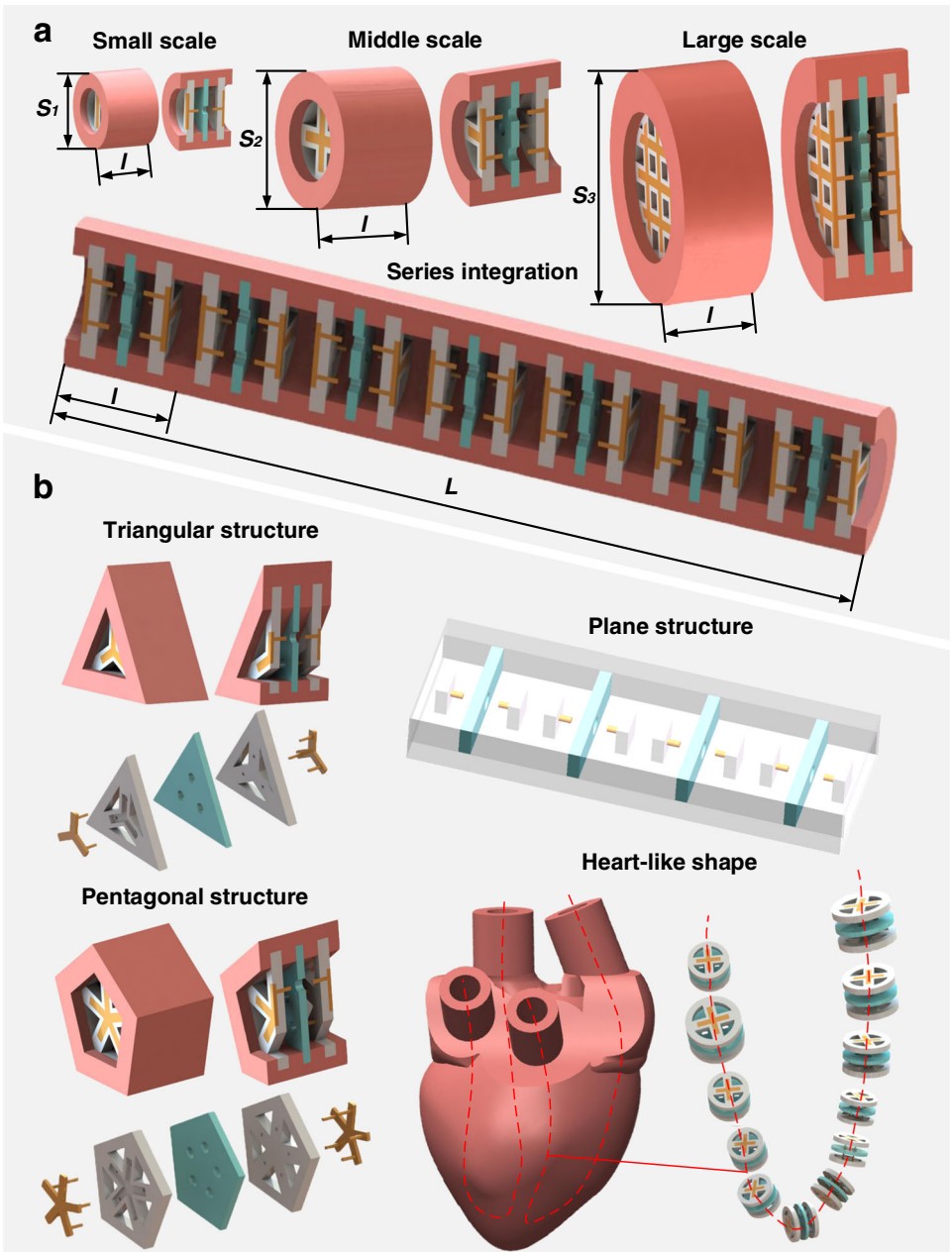

**Fig. 2 Schematic illustration of customizable appearances of soft electronic pumps. a** Various sizes of soft electronic pumps. The multiscale circumferential cross-sectional areas ($S_1$, $S_2$, $S_3$) could be achieved by combining the different number of minimum electrode pair modular unit, while the multiscale axial length ($L$) can be achieved by integrating the different number of the pump. **b** Various shapes of soft electronic pumps. Different shapes, including triangular structure, pentagonal structure, plane structure, and complicated heart-like shape, could be achieved by changing electrode layout, support shape, and shell shape.

We compared the pumpabilities of soft electronic pumps with several commercially available pumps and compressors widely used to power soft robots. A soft pump integrating nine needle-hole electrode pairs can generate a pressure greater than 60 kPa and a flow rate higher than 350 ml/min, comparable to three commercial pumps with the same diameter and length (Fig. 3g and Supplementary Table 3). The lightweight (3 g) miniature (0.68-cm thickness) soft electronic pump can generate a pressure of ~9.2 kPa and a flow rate of ~423 ml/min. The maximum power consumption of a soft electronic pump is ~3.6 W (voltage ~16 kV, electric current ~225 μA). The soft electronic pumps merge the unique properties of lightweight and powerful output, enabling them to have larger specific pressures (~3066.67 kPa kg$^{-1}$) and

specific flow rates (~141,000 ml min$^{-1}$ kg$^{-1}$) over those commercial pumps/compressors and human heart (Fig. 3g).

Relative to other existing stimulus-responsive fluid driving power sources resulted from combustion[17], phase transition[18], and stretchable electrohydrodynamic pumps[23], etc., our soft electronic pumps display excellent comprehensive performances in powerful actuation, rapid response, excellent portability, good controllability, long lifetime, high robustness, diverse, easy fabrication, and low cost. Compared with current soft combustion-driven pumps[17], the soft electronic pump integrating nine needle-hole electrode pairs in series could achieve the same pressure (~60 kPa), but the flow rate of soft electronic pump (~350 ml min$^{-1}$) is much higher than those of soft combustion-driven pumps (~40 ml min$^{-1}$). Meanwhile, soft

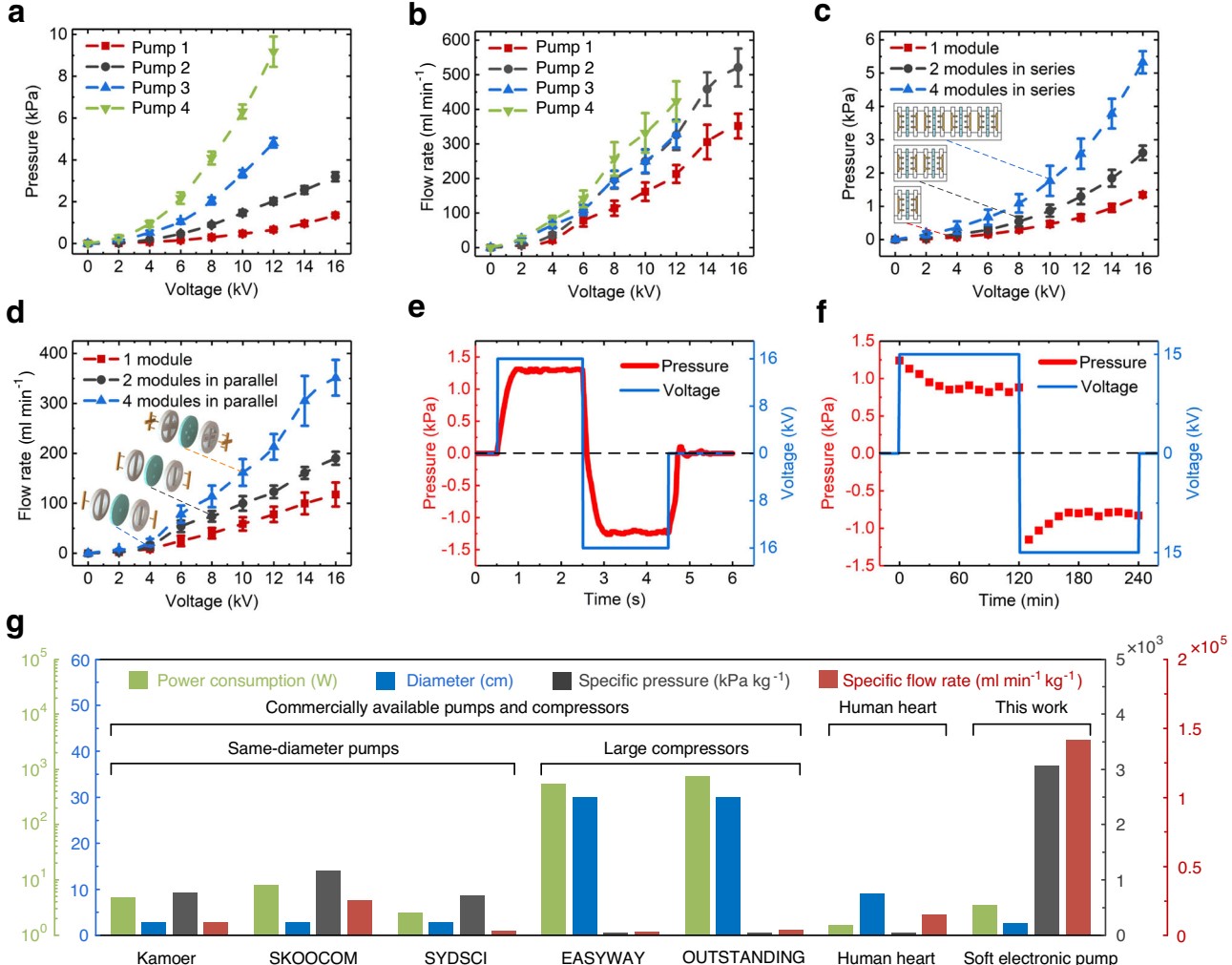

**Fig. 3 Customizable pumpabilities of soft electronic pumps. a** Generated pressure-voltage curve of different combinations of electrode designs and liquids. Pump 1: Electrode design 1 + Liquid 1; Pump 2: Electrode design 1 + Liquid 2; Pump 3: Electrode design 2 + Liquid 1; Pump 4: Electrode design 2 + Liquid 2. Where, Electrode design 1: 1-mm diameter needle, 3-mm diameter hole, and the 2-mm gap between them; Electrode design 2: 0.4-mm diameter needle, 1-mm diameter hole, and 0.8-mm gap between them; Liquid 1: Dibutyl sebacate based functional liquid; Liquid 2: Linalyl acetate-based functional liquid. A clearer description of the four pumps can be seen in Supplementary Table 2. The dielectric breakdown voltages of pump 1 and pump 2 are above 16 kV, while those of pump 3 and pump 4 are ~14 kV. **b** Flow rate-voltage curve of different combinations of electrode designs and liquids. **c** Influence of series integration of multiple needle-hole electrode pairs on the generated pressures. **d** Influence of parallel integration of multiple needle-hole electrode pairs on the flow rates. **e** Switching response time of the soft electronic pumps. **f** Lifetime test of soft electronic pumps. **g** Performance comparison between the soft electronic pump, various commercially available pumps/compressors (Kamoer, EDLP600; SKOOCOM, SC3711PW; SYDSCI, DLP100-DC; EASYWAY, E8L-550W; OUTSTANDING, 750-30L) and a human heart, the details of the commercially available pumps/compressors can be seen in Supplementary Table 3. Pump 1 was used for **c-f**: Series/parallel use, response speed, and life are studied. In **g**, the most powerful Pump 4 was used.

electronic pumps are controllable, long-term, and robust. The response speeds of soft pumps are much faster than existing thermal-activated phase transition driving power sources[18]. Compared with stretchable electrohydrodynamic pumps[23], needle-hole electrode configuration of soft electronic pumps not only could generate much more powerful pressures and flow rates, but also benefits for flexible and diverse spatial arrangement, overcoming the shortcomings of insufficient output capacities, slow system responses, and monotonous appearances of stretchable electrohydrodynamic pumps. More detailed comparisons of the soft electronic pump and stretchable electrohydrodynamic pump are indicated in Supplementary Table 1. Hence, the results of tests and comparison with existing fluid power technologies demonstrate that soft electronic pumps are suitable to be a universal driving power source for driving various soft robotic systems.

**High-voltage power converter for soft electronic pumps**. In order to facilitate the implantation of the soft electronic pump into diversified soft robots, we design a lightweight and miniature high-voltage power converter (HVPC) (Supplementary Fig. 10a–c) that can convert low-voltage input of battery or wireless power transfer system to high-voltage output for powering soft electronic pumps. Particularly, our HVPC can simultaneously offer two independent high-voltage outputs, allowing for the reversible bidirectional pumping of soft electronic pumps. Supplementary Fig. 10d and Supplementary Video 4 demonstrate the rapid bidirectional motion of a soft actuator implanted with a soft electronic pump under the power supply of HVPC, and the response time of the actuator is ~1 s, which is comparable to that of traditional hydraulic and pneumatic systems.

**Untethered robots powered by soft electronic pumps**. We illustrate the wide potential of soft electronic pumps for next-generation built-in driving power sources of robotic systems by demonstrating the untethered and versatile motions of different robots powered by the pumps. When implanted into a bionic soft robotic fish (Fig. 4a), the circular soft electronic pump can drive a bidirectional two-chamber bending actuator that is capable of mimicking vividly the swing of the tail fin to realize the untethered underwater propulsion of soft robotic fish. The detailed working principle of the tail's bidirectional bending motion is that continuous bidirectional bending motion can be achieved by pumping the liquid back and forth in both two chambers when a soft electronic pump is connected to the chambers of the tail through channels. Miniature HVPC with a power supply of a lithium battery is also embedded in the fish body. The rapid bidirectional bending motion of soft tail fin in the air is shown in Fig. 4b and Supplementary Video 5, while the silent and untethered swimming motion underwater is shown in Fig. 4c and Supplementary Video 5. When implanted into a robotic vehicle (Fig. 4d), the square soft electronic pump can drive a linear actuator to push the robotic vehicle forward. The HVPC with a power supply by two coils that can wireless receive energy emitted by coils laid on the bottom are embedded in the robotic vehicle (Fig. 4e). The movement of the robotic vehicle is realized in two steps. The first step is the elongation of the linear actuators, while the second step is the contraction of the actuator. In the elongation process ($T_1$–$T_2$), the two brackets move away from each other. The front wheels roll forward while the rear wheel is unable to roll back due to the one-way bearing, providing forward friction to the robotic vehicle. In the contraction process ($T_2$–$T_3$), the two brackets move close to each other. The rear wheels roll forward and the front wheel provides forward friction to the robotic vehicle. Repeating these two steps circularly, the robotic vehicle can continuously move forward powered by the wireless power (Fig. 4f and Supplementary Video 6). Supplementary Fig. 11c shows the self-healing processes of the robotic vehicle.

## Discussion

In this study, we report bio-inspired soft electronic pumps as driving power sources to actuate and heal untethered soft robots. These soft electronic pumps offer several promising qualities: fully soft architecture, powerful actuation, rapid response, long lifetime, good controllability, excellent portability, high robustness, versatility, self-healing, easy fabrication, and low-cost. Soft electronic pumps combined with our homemade lightweight and miniature HVPCs could be fully implanted into most existing soft robots[2,8–10] to get rid of the tethers and become free, representing a tremendous progress in the development of untethered soft robotics.

## Methods

**Materials and fabrication of soft electronic pumps**. Soft electronic pumps consisted of four basic components: two needle electrodes, a ring electrode, two electrode supports, and a shell. The fabrication process of a soft electronic pump mainly included four steps (Supplementary Fig. 1): (i) Step 1–Fabrication of needle electrode, (ii) Step 2–Fabrication of ring electrode, (iii) Step 3–Fabrication of shell, and (iv) Step 4–Assembly of all components. (i) Step 1 (Supplementary Fig. 1–Step 1): The needle electrode was made of conductive silicone 50A (BD-7160, China Hangzhou Bald Advanced Materials Co., Ltd.) and silicone 30A (PS6600-5du and Translucency, China Shenzhen Yipin Trading Co., Ltd.). The needle electrode was fabricated using silicone casting and coating. Silicone 30A was prepared by mixing parts A and B in a 1:1 ratio then degassing them in a vacuum dryer for about 3 min. Uncured silicone 30A was cast into the mold of the needle and cured for ~6 h at room temperature, then removing the mold to form the needle support. Conductive silicone 50A was prepared by mixing parts A and B in a 10:1 ratio then degassing them in a vacuum dryer for about 3 min. Uncured conductive silicone 50A was coated in all surfaces of the needle support and cured for ~5 h at 90 °C to form a conductive needle. The conductive needle was then placed in the mold of the electrode, making sure their relative position is correct. Uncured silicone 30A

was cast into the mold and cured for ~6 h at room temperature, then removing the mold to form the needle electrode. Finally, an enameled wire was connected to the needle electrode, and the resistance of the needle electrode was ~5–20 kΩ. (ii) Step 2 (Supplementary Fig. 1–Step 2): The ring electrode was made of commercially available conductive silicone 60A sheet (2-mm sheet/1-mm sheet, Dongguan Ziming Silicone Rubber Co., Ltd.). The conductive silicone 60A sheet was cut into a given shape to form a ring electrode, and then an enameled wire was connected to the ring electrode. The resistance of the ring electrode was ~200–300 Ω. (iii) Step 3 (Supplementary Fig. 1–Step 3): The support was made of silicone 5A (PS6600-5du and Translucency, China Shenzhen Yipin Trading Co., Ltd.) or polydimethylsiloxane (PDMS) (Sylgard 184, Dow-Corning). Silicone 5A was prepared by mixing parts A and B in a 1:1 ratio then degassing them in a vacuum dryer for about 3 min. Uncured silicone 5A was cast into the molds of the shell and cured for ~3 h at room temperature, then removing the molds to form the shell. PDMS was prepared by mixing parts A and B in a 10:1 ratio then degassing them in a vacuum dryer for about 3 min. Uncured PDMS was cast into the molds of the shell and cured for ~5 h at 90 °C, then removing the molds to form the transparent shell. (iv) Step 4 (Supplementary Fig. 1–Step 4 and Supplementary Video 1): two needle electrodes and one ring electrode were installed in the shell to form the soft electrode pump, where each needle in needle electrodes was aligned with the center of each hole in the ring electrode. All parts of the soft electronic pump were soft and stretchable.

**Liquids pumped by soft electronic pumps**. Three kinds of liquids, including dibutyl sebacate (D108607, Aladdin), linalyl acetate (L101415, Aladdin), and tung oil, were used for soft electronic pumps. Dibutyl sebacate, as commonly used as packaging material in contact with food, a raw material of spice, a cold-resistant plasticizer, and a gas chromatography stationary liquid, was commercially available. Linalyl acetate, a kind of spices, was widely used in the preparation of soap flavors, perfume flavors, cosmetic flavors, food flavors, and so on. Tung oil is an excellent drying oil of vegetable origin used principally in the preparation of paints, varnishes, and related materials. Dibutyl sebacate-tung oil solution was formed by dissolving tung oil in dibutyl sebacate, where their volume ratio is 1: 2, that is the self-healing liquid. A set number of self-healing droplets were attached to silicone sheets, and a small glass rod was used to touch the self-healing area every hour to evaluate whether the self-healing liquid is solidified. At the same time, a timer was used to record the self-healing time. The self-healing time of the liquid on the silicone sheet was ~6 h at 35 °C and ~1 day at 24 °C. The self-healing liquid sealed in the elastomer chamber could still maintain its pumping and self-healing performance after about two months.

Viscoelastic properties of the self-healed films were measured by dynamic thermomechanical analysis (DMA) tests using a thermodynamic analyzer (RSA-G2, TA), as shown in Supplementary Fig. 5. The thickness and width of the self-healed film were 1 mm and 10 mm, respectively. The duration, oscillation strain, and frequency of the DMA test were set as 60 s, 5%, and 10 Hz, respectively. To evaluate the reliability, accuracy, and repeatability of experimental results, we tested three samples and each sample was tested three times. The DMA results are shown in Supplementary Fig. 5. It can be seen from the figure that the self-healed film is transparent and elastic. The storage modulus of the self-healed film is ~14.3 kPa at 20 °C, ~13.2 kPa at 25 °C, and ~10.6 kPa at 30 °C, respectively. The loss modulus of the self-healed film is ~ −3.1 kPa at 20 °C, ~ −2.5 kPa at 25 °C, and ~ −2.1 kPa at 30 °C, respectively. It is apparent that the storage modulus and loss modulus decreased as the temperature increased. To further characterize the adhesion property between the self-healed film and the silicone film, we conducted the repeated tensile tests of a silicone film with local self-healed film, as shown in Supplementary Fig. 6a. The self-healing liquid was dropped into a 1-mm-thickness silicone sheet and then solidified to form a self-healed film bonding with the silicone sheet. The silicone sheet bonding with a self-healed film was cut into a size of ~30 mm × 10 mm × 1 mm and then was used to conduct a tensile test (Universal tensile testing machine, ZQ-990B, Dongguan Zhiqu precision instruments Co., Ltd). In the tests, the silicone film was repeatedly stretched 200 times with 20% strain per time. After tests, it was found that the silicone film and the self-healed film were still firmly bonded together, illustrating good adhesion property between the two films. To compare the mechanical properties of pristine and self-healed samples, uniaxial tensile tests were conducted at an ASTM D638 (Type IV) universal test machine with a crosshead speed of 50 mm/min. Supplementary Fig. 6b showed that the uniaxial tensile stress-strain curves of pristine and self-healed samples are close, illustrating that the difference in the mechanical properties between them is small. Note that the applied stresses of the self-healing sample are slightly greater than those of the pristine sample to achieve the same strain due to that the local self-healing film makes the self-healing sample slightly harder than the pristine sample.

**Theoretical model and numerical simulation of the soft electronic pump**. To obtain distributions of electric potential, charge density, and velocity in the soft electronic pumps, we develop theoretical model[24,25] and numerical simulation. The equations that describe the flow, electric field, and current density are given below. Two-dimensional and steady flow is assumed.

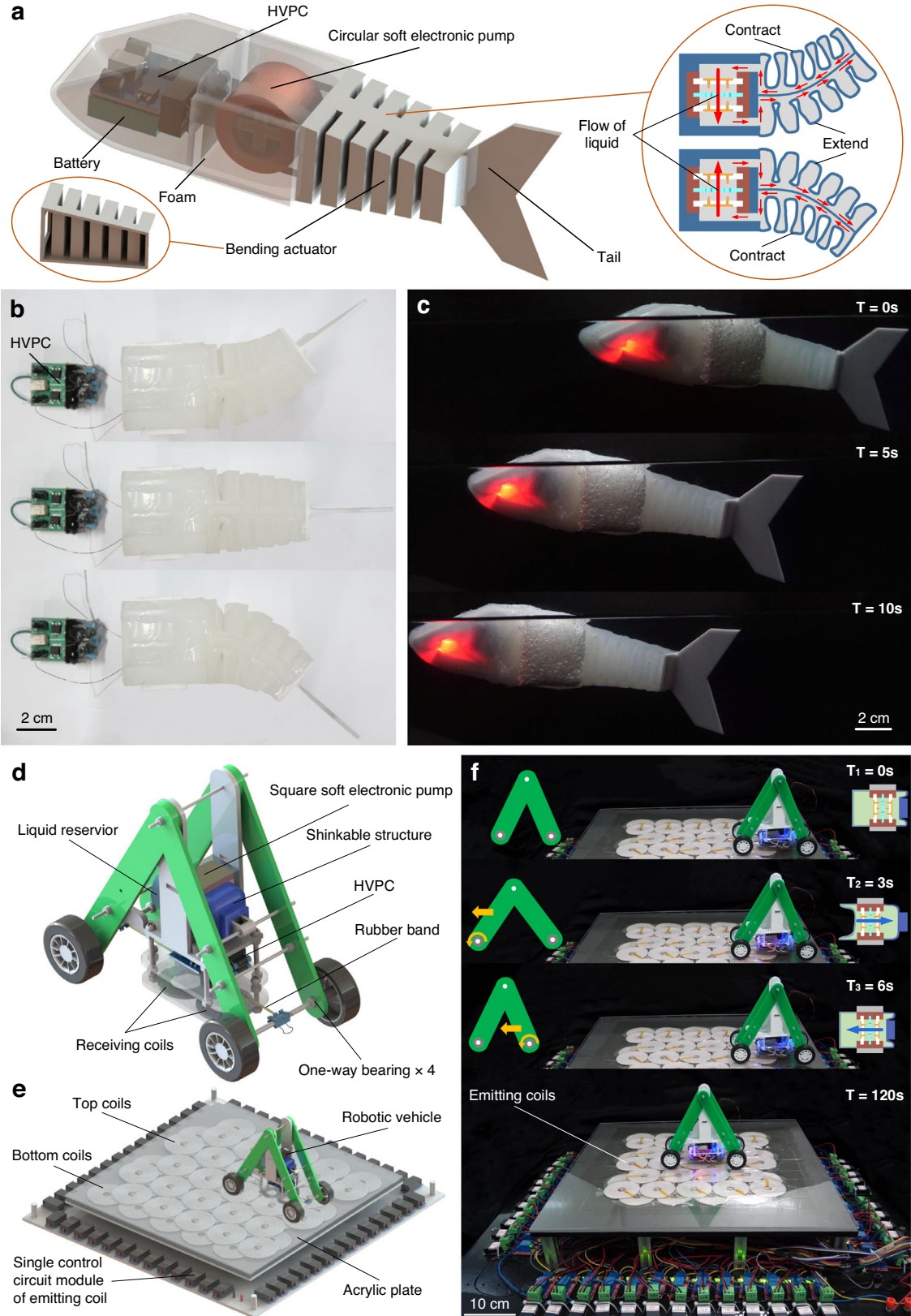

**Fig. 4 Untethered robots powered by soft electronic pumps. a** Design and operational principle of soft robotic fish. A circular soft electronic pump is implanted into the robotic fish to achieve the swing of the tail fin. **b** Tail fin actuation in air. **c** Time-lapse of the untethered swimming motion underwater of the fish. **d** Computer-generated rendering of the robotic vehicle. A square soft electronic pump is implanted into the robotic vehicle to achieve its forward motion. **e** Schematic of the robotic vehicle powered by the wireless power transfer system. The emitting coils are closely arranged in two layers, and the on/off of the emitting coils is determined according to the position at which the robotic vehicle is at this moment and the next moment. **f** Time-lapse of the robotic vehicle moving on the top of the emitting coils. Schematics of the robotic vehicle are inserted in the upper left and right corner of each picture.

Because the flow is incompressible, the continuity equation is given as:

$$\nabla \cdot \mathbf{u} = 0 \tag{1}$$

where $\mathbf{u}$ is the flow velocity.

For stead flow, the Navier-Stokes' equation is given as:

$$(\mathbf{u} \cdot \nabla)\mathbf{u} = \mathbf{f} - \frac{1}{\rho}\nabla p + \frac{\eta}{\rho}\nabla^2 \mathbf{u} \tag{2}$$

where $\mathbf{f}$ is the potential force, $p$ is the pressure, $\rho$ is the liquid density, and $\eta$ is the liquid viscosity.

The Coulomb force $\mathbf{f}$ that works on the liquid is given as:

$$\mathbf{f} = \frac{1}{\rho}q\mathbf{E} \tag{3}$$

where $q$ is the charge density, and $\mathbf{E}$ is the electric field.

According to Gauss' law, the electric-potential distribution in liquid is given as:

$$\nabla^2 \phi = -\frac{q}{\varepsilon_0 \varepsilon_r} \tag{4}$$

where $\phi$ is the electric potential, $\varepsilon_0$ is the permittivity of free space, and $\varepsilon_r$ is the relative permittivity of liquid.

The electric-field distribution can be calculated as:

$$E = -\nabla \phi \tag{5}$$

The current density in liquid is calculated as:

$$\mathbf{j} = q\mu_i \mathbf{E} + q\mathbf{u} - D_i\nabla q + \sigma \mathbf{E} \tag{6}$$

where $\mathbf{j}$ is the current density, $\mu_i$ is the ion mobility, $D_i$ is the charge diffusion coefficient, and $\sigma$ is the conductivity of the liquid.

The charge conservation law is given as:

$$\nabla \cdot \mathbf{j} = 0 \tag{7}$$

Charge density in positively charged process is determined as:

$$q_e = k(E_{\text{static}} - E_{\text{thres}}) \tag{8}$$

where $k$ is the proportional constant for injected charge density, $E_{\text{static}}$ is the electric field of the electrode surface, and $E_{\text{thres}}$ is the threshold value of the electric field of the electrode surface to flow generation.

The boundary conditions for the soft electronic pump are given to the surfaces (1) to (6) shown in Supplementary Fig. 4a as follows:

$$(1) \qquad u = U_{in}, \ v = 0, \ \frac{\partial \phi}{\partial x} = 0 \tag{9}$$

$$(2) \qquad \frac{\partial u}{\partial x} = 0, \ \frac{\partial v}{\partial x} = 0, \ \frac{\partial q}{\partial x} = 0, \ \frac{\partial \phi}{\partial x} = 0 \tag{10}$$

$$(3) \qquad u = 0, \ v = 0, \ \frac{\partial \phi}{\partial y} = 0 \tag{11}$$

$$(4) \qquad u = 0, \ v = 0, \ q = q_e, \ \phi = V \tag{12}$$

$$(5) \qquad u = 0, \ v = 0, \ \phi = V \tag{13}$$

$$(6) \qquad u = 0, \ v = 0, \ \phi = 0 \tag{14}$$

The theoretical model described above is solved using the finite-element-method based on commercial software COMSOL Multiphysics. A needle-hole electrode pair of pump 1 is used to numerically simulate and validate the theoretical model. The liquid (dibutyl sebacate-based functional liquid) properties used in the simulation are listed in Table 1.

The density, viscosity, relative permittivity, and conductivity of dibutyl sebacate-based functional liquid were tested based on our experimental apparatus. The density, viscosity, relative permittivity, and conductivity of dibutyl sebacate based functional liquid were ~938 kg m$^{-3}$ (WLD-300A, Electronic density tester, Wanliduo), $7.5 \times 10^{-3}$ Pa s (NDJ-5S-5T, Digital rotary viscometer, Shanghai Fangrui Instrument Co. Ltd.), ~4.5 (Vector network analyzer, coaxial reflection method, measurement frequency is 1 GHz) ~$3.4 \times 10^{-10}$ S m$^{-1}$ (Liquid Conductivity Meter), respectively. The ionic mobility of dibutyl sebacate-based functional liquid is obtained based on the method used in the paper[25] (where dibutyl decanedioate and dibutyl sebacate are the same liquids) and lots of result comparisons of experiments and simulations. The ionic mobility of dibutyl sebacate-based functional liquid is ~$6.93 \times 10^{-8}$ m$^2$ V$^{-1}$ s$^{-1}$.

Supplementary Fig. 4b–d shows the distributions of electric potential, charge density, and velocity. The calculated flow rate resulted from the simulation is ~339 ml min$^{-1}$ at 16 kV, while the experimental flow rate of prototype pump 1 is ~351 ml min$^{-1}$ at 16 kV. The error between the experimental value and the calculated value is ~3.4%, demonstrating that the theoretical model and numerical simulation of the soft electronic pump are reliable and correct.

**Table 1 Physical properties of dibutyl sebacate-based functional liquid.**

| Item | Value |
|---|---|
| Density $\rho$ (kg m$^{-3}$) | 938 |
| Viscosity $\eta$ (Pa s) | $7.5 \times 10^{-3}$ |
| Relative permittivity $\varepsilon_r$ | 4.5 |
| Conductivity $\sigma$ (S m$^{-1}$) | $3.4 \times 10^{-10}$ |
| Ionic mobility $\mu_i$ (m$^2$ V$^{-1}$ s$^{-1}$) | $6.93 \times 10^{-8}$ |

**Experimental procedures of performance tests**. A custom LabVIEW program (Version 2017, 32-bit), a data acquisition board (Model USB-6341, National Instruments), and a high voltage amplifier (Model 20/20C, Trek) were used to generate the high voltage signals for the performance tests of soft electronic pumps, and an oscilloscope (InfiniiVision 2000 X-Series, Keysight) was used to display the high voltage signals. The experimental setups for measuring generated pressures and flow rates were shown in Supplementary Fig. 6a, b, respectively. The measuring system for generated pressures consisted of a liquid reservoir, tubes, a soft electronic pump connected to the high voltage amplifier, a manometer, and an ammeter. The generated pressures were measured by the manometer, and the electric currents were measured by the ammeter. To prevent the effect of a pressure difference caused by a height difference, the liquid level in the reservoir, the position of the soft electronic pump, and the height of the manometer were kept at the same height. The measuring system for flow rates consisted of a liquid reservoir, tubes, a soft electronic pump, a graduated cylinder, an electronic timer, and an ammeter. The flow rates were calculated from the volume measured by the graduated cylinder within a certain period of time. In the system, the reservoir, the soft electronic pump, and the tube were also kept at the same level to measure the flow rate without load pressure. For each version of the soft electronic pump, three same pumps were fabricated and each pump was tested three times. Thus, a total of nine values were obtained for each pump. Finally, the average value and standard deviation were calculated based on the nine values to evaluate the reliability, accuracy, and repeatability of experimental results.

**High-voltage power converter**. The lightweight and miniature two-output high-voltage power converter (HVPC) was designed and fabricated to power the soft electronic pumps, thereby achieving fast-switching bidirectional pumping when the pumps were implanted into robotic systems. The two-output HVPC included two DC HVPC (Supplementary Fig. 10a). The DC HVPC adopted a conventional topology that consisted of a pulse oscillating circuit, inverter circuit, high-frequency transformer, and voltage doubling rectifying circuit[29,30]. The pulse oscillating circuit was mainly made of a timer NE555 to output a pulse of certain pulse width for driving the MOSFET. The inverter circuit was mainly made of a MOSFET IRF540N to form a flyback topology drive circuit, converting the input DC power into a high-frequency square wave pulse. After that, the high-frequency square wave pulse was boosted by a high-frequency transformer and then converted into a required DC high voltage by a voltage doubler rectifier circuit. The two-output HVPC mainly consisted of two DC HVPC and a control circuit. The control circuit was used to control the two DC HVPC, realizing one DC HVPC on/off and the other DC HVPC off/on. To reduce electromagnetic interference, the two transformers were separated by a non-magnetic shim. A high-voltage probe (P6015, Trek) and an oscilloscope (InfiniiVision 2000 X-Series, Keysight) were used to test the output of the HVPC, and the results showed in Supplementary Fig. 10c.

**The wireless power transfer system**. The wireless power transfer[31,32] system was mainly composed of 50 emitting circuit modules, 50 emitting coils, 2 receiving circuit modules, 2 receiving coils, a non-magnetic shim, 50 relays, a power supply, a microcontroller, and a two-layer nylon plate frame (Fig. 4f and Supplementary Video 6). The emitting coils were closely arranged in two layers, where 27 coils were in the bottom layer and 23 coils were in the top layer. All emitting coils were mounted to the non-magnetic shim using double-sided tape. The diameter of the emitting coil was 70 cm, and the receiving coil was 35 cm. The effective power supply distance of wireless power transfer was about 15 mm, the output voltage was about 7 V, and the output electric current was about 1.5 A. The microcontroller and relays were used to control the emitting circuit modules on and off.

**Bidirectional pumping between two liquid reservoirs**. To demonstrate the rapid, controllable, and fast-switching bidirectional pumping capacity of the soft electronic pump, the pump was connected to the two cylindrical liquid reservoirs with connectors and tubes so as to move liquid between them. The inner diameter of the cylindrical liquid reservoir and tube was 54 mm and 18 mm, respectively. The liquid heights in two cylindrical liquid reservoirs were ~20 mm. Square waves with different amplitudes were applied to power the soft electronic pump, as shown in Supplementary Fig. 3 and Supplementary Video 2. The pumping time between the

two cylindrical liquid reservoirs was ~3.1 s, 8.5 s, and 6.5 s, respectively, and the total time was ~18.1 s under 16 kV. The total pumping time under 12 kV and 8 kV was ~28.1 s and ~45.1 s, respectively.

**Soft bidirectional actuator**. The soft bidirectional actuator (Supplementary Fig. 10d) consisted of a soft electronic pump, two shrinkable structures and support. The shrinkable structure and support were made of Ecoflex 00-20 (Smooth-on) and silicone 5A, respectively, and fabricated by mould-casting. These parts were aligned and bonded together by applying uncured silicone 5A at their interface and cured for ~3 h at room temperature to form the body of the actuator. About 16-mL liquid was then filled into the chamber by using a syringe with a needle to form the self-contained soft bidirectional actuator. Supplementary Fig. 10d and Supplementary Video 4 demonstrate the rapid bidirectional motion of the soft actuator under the power supply of HVPC, and the response time of the actuator is ~1 s, which is comparable to that of traditional hydraulic and pneumatic systems.

**Soft robotic fish**. The soft robotic fish (Fig. 4a) consisted of a tail fin implanted into a circular soft electronic pump, a body, two foams, an HVPC, and a battery. The tail fin consisted of a bidirectional two-chamber bending actuator. The bending actuator was made of Ecoflex 00-30 (Smooth-on) and fabricated by mould-casting. About 25-mL liquid was filled into the chamber by using a syringe with a needle to form the self-contained soft bidirectional bending actuator. The rapid bidirectional bending motion of soft tail fin in the air is shown in Fig. 4b and Supplementary Video 5, while the silent and untethered swimming motion underwater is shown in Fig. 4c and Supplementary Video 5.

**Robotic vehicle**. The robotic vehicle (Fig. 4d and Supplementary Fig. 11b) consisted of a linear actuator implanted into a square soft electronic pump, two brackets, a rubber band, four wheels, four one-way bearings, an HVPC, and two receiving coils. The linear actuator (Supplementary Fig. 11a) consisted of a square soft electronic pump, a shrinkable structure, a liquid reservoir, support, and a connector. The shrinkable structure, liquid reservoir, and support were made of Ecoflex 00-30 (Smooth-on), Ecoflex 00-20, and silicone 5A, respectively, and fabricated by mould-casting. About 15-mL liquid was filled into the chamber by using a syringe with a needle to form the self-contained soft linear actuator. To make sure that the robotic vehicle has a receiving coil at one of the emitting coils at any moment of motion, the emitting coils are closely arranged in two layers and the diameter of the emitting coil is twice that of the receiving coil. Figure 4f and Supplementary Video 6 show the forward motion of the robotic vehicle powered by the wireless power transfer system. Supplementary Fig. 11c shows the self-healing processes of the robotic vehicle.

## Data availability
The data that support the findings of this study are available from the corresponding authors upon reasonable request.

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

## Acknowledgements
This work is supported by the National Natural Science Foundation of China (award number 51875507 and 51890885), and the Fundamental Research Funds for the Central Universities.

## Author contributions
W.T., C.Z., H.Y., and J.Z. conceived the idea. W.T., C.Z., and J.Z. designed the research. W.T., C.Z., Y.Z., P.Z., Y.H., Z.J., X.W., G.L., J.W., Y.W.L., Y.Q.L., and W.W. conducted the experimental work. W.T., C.Z., and J.Z. designed and fabricated the high-voltage power converter and the wireless power transfer system. W.T., C.Z., and J.Z. analyzed the data. W.T., C.Z., and J.Z. wrote the manuscript with input from all authors.

## Competing interests
The authors declare no competing interests.
