## [Peer Review File · Nature Communications]

Reviewer #1 (Remarks to the Author):

This manuscript reports a customizable modular soft pump that demonstrates specific pressure and specific flow rate superior to commercial pumps and other soft pumps. The enabling design feature is the needle-hole electrode pair configuration. The author also proposes self-healing capabilities by choosing fluids that can solidify when brought in contact with air, and draws bioinspired comparison with spiders. At last, the author demonstrates wireless pumping capabilities in a robotic fish and a robotic vehicle.

The manuscript proposes a technology that is potentially generalizable to soft robotic systems, but more data need to be presented. It is an exciting project, but missing much of the information that would qualify it for publication. Therefore, I would recommend acceptance given the following revisions:

1. The author claimed self-healing as a main feature of the pump and explained the working principles. There is, however, no evidence in the form of data or photo, that the self-healing function proposed is realized. The figures only show schematics of working principles. I suggest the author to present data that support the self-healing claim.
2. One major issue is the writing. While this work presents some good results such as the pump performance, the writing is structured in an unorganized way that is hard to follow.
 - a. The abstract and introduction does not highlight the contribution of this paper. The author tries to present the pump to be advantageous in every way (“fully soft, powerful, controllable, rapid response, built-in, long-term, and self-healing”) but the reader is lost in the real highlight of the contribution.
 - b. In the introduction, the author is too generalized in the background introduction by saying every mechanism has their unique disadvantage. A good literature review such as later discussed in line 222-241 should be summarized at the introduction to give readers an idea of the problem this paper is addressing.
 - c. The bioinspiration from spiders could make sense if the author is more clear about what is the similarity. Is the needle-hole structure inspired by the spider’s heart? Or is the pump mimicking the functions of pumping and self-healing?
 - d. There needs to be a design section after the working principle is introduced to explain better each parameter’s effect on the pumping performance of a single module. The author has this study but it’s buried later in the text. I suggest designating a paragraph that discuss the findings in Supplementary Fig. 7.
 - e. There are lots of redundancy throughout the writing. Such as line 30-30.
3. Fig. 2 are all schematics. I suggest bringing the photos of design variations to Fig. 2.
4. Fig. 3b has a mismatching color for pump 1.
5. Are there self-healing demonstrations in the robotic fish and vehicle? The conclusion seems to suggest that but I haven’t found it in the robotic demonstrations.
6. More detailed improvements are needed here:
 - Page 5, figure 1e and page 7, line 107: In figure 1e, the authors demonstrated the jet flow caused by electrical field, and only showed positive ions in the liquid. The authors further explained in page 7 that the mechanism of ion generation is through “positively charged process”. However, for different kinds of functional fluids, the dominant ion species could be either positive or negative. For example, in figure 1b from reference [15], negatively charged ions were the dominant species (Cacucciolo, V. et al. Stretchable pumps for soft machines. *Nature* 572, 516–519 (2019).). Ref [25] reported a way to distinguish dominant species via Sumoto effect (Kurobosh, Y., Takemura, K. & Edamura, K. Understanding of electro-conjugate fluid flow with dibutyl decanedioate using numerical simulation—Calculating ion mobility using molecular dynamics simulation. *Sens. Actuators*

B Chem. 255, 448–453 (2018).). The reviewer would like to know if figure 1e is valid for both functional fluids (i.e., dibutyl sebacate and linalyl acetate) used in this paper?

- Page 8, line 152: the authors claimed that it took ~ 6 hours to achieve complete self-healing process under ambient temperature and the healed system could work for another 2 hours without leakage. Since self-healing is achieved by solidification of tung oil, which has a different composition from silicone, the reviewer would like to see a more quantitative way to characterize mechanical property of the pump.
- Page 10, line 180: supplementary fig. 6 is not called out throughout the main text. It should be placed somewhere in this paragraph.
- Page 11, line 223 and page 13 figure 3a: the authors mentioned that the soft electronic pumps could reach similar pressure (~ 60 kPa) to current soft combustion-driven pumps. However, there isn't enough experimental evidence indicating that the soft pump can reach ~ 60 kPa. In figure 1a, the highest pressure on the plot is below 10 kPa. Furthermore, the data of Pump 3 and Pump 4 under 14 kV and 16 kV are missing. The reviewer would like to know the reason.
- Page 13, figure 3b: the line color of Pump 1 is inconsistent with the one used in figure 3a.
- Page 22, line 451: Please use a gradient operator.
- Page 24, line 473: Please specify the data source of these physical properties.
- Since one important design feature of this pump is that it can operate bidirectionally, as it was used in the robotic fish, the reviewer would like to see if the cyclability of this kind of pump. Moreover, the authors should also briefly discuss the frequency limits of this pump under AC signal.

Reviewer #2 (Remarks to the Author):

Summary

This manuscript (Customizing a self-healing, soft pump for robot) shows an electrohydrodynamic pump for soft robots. The pump contains a soft structure that contains a liquid dielectric layer and soft electrode. When a high voltage is applied between the electrodes, ions move from one electrode to another, and drag fluid to create a pumping effect. The pump is also able to self-heal from mechanical damage, as the liquid would slowly form a thin film when exposed to air. Furthermore, the authors showed applications as an untethered robotic fish and an untethered vehicle, and they stated the pump has promising qualities, such as fully soft architecture, powerful actuation, rapid response, long lifetime, good controllability, excellent portability, high robustness, versatility, self-healing, easy fabrication, and low-cost. Under my point of view, the data and results shown in this paper could further be improved, and there are some comments that the authors need to address. Thus, I believe this paper cannot be accepted in Nature Communications until more explanations are included in the corrected version of the submitted manuscript.

Comments to the Authors

1. In the introduction, the authors referenced electrostatic force (ref. 15) have their own distant

shortcomings when used in untethered soft systems. The authors should remove the electrostatic force, as this paper's mechanism is still based on electrostatic force mechanism, as the electric field causes positive ion migration.

2. To make the introduction more convincing, the authors should provide details on what they could do better than ref. 15 in the introduction. Many of these examples could be seen in Table 1, in which the authors provided for previous reviewer 1 for Nature Materials submission (such as Positive ions migration is higher than negative ions, much higher pressure of single electrode pair, less liquid dragging in the liquid flow)

3. In the second paragraph of introductions, more detail is needed on how the haemolymph works, such as explaining how the haemolymph could generate hydraulic pressure. I suggest the authors to provide more detail and also to cite Figure 1a in the introduction section.

4. In line 120-122, the authors mentioned "Such free electrons will be absorbed into needle positive electrodes from liquid, resulting in formation of free ions with positive charge." It is difficult to visualize how these electrons absorbing into positive electrodes would form positive charged free ions. Please provide more details on how this positive charged free ions are formed.

5. For self-healing, the authors states that it takes 6 hours at an ambient temperature (35 degrees C) for the self-healing process. However, 35 degrees C is not ambient temperature. I suggest the authors to perform the test at room temperature.

6. Also, since the self-healing of the tung oil is a novelty of this work. I suggest the authors to do more characterization of the self-healed film, such as DMA tests at different time and temperature of self-healing.

7. The authors showed Figure 3a, but did not mention what are the four pumps being tested in the manuscript. The author should clarify by adding a table in the manuscript showing the difference of each pump that is used.

8. The authors showed in Figure 3f showing that applying a voltage of 15kV for 4 hours, showing the durability and reliability of soft pumps. It could be seen that the pressure starts to decrease when holding the constant voltage to longer times. I suggest the authors should provide an explanation for why the pressure would decrease after applying steady voltage.

9. In Figure 3g, the authors showed the power consumption of their work to be around 3-4 W. The authors should provide calculations on how they were able to obtain this value.

Detailed Responses to Reviewers

Reviewer #1 (Remarks to the Author):

In summary: This manuscript reports a customizable modular soft pump that demonstrates specific pressure and specific flow rate superior to commercial pumps and other soft pumps. The enabling design feature is the needle-hole electrode pair configuration. The author also proposes self-healing capabilities by choosing fluids that can solidify when brought in contact with air, and draws bioinspired comparison with spiders. At last, the author demonstrates wireless pumping capabilities in a robotic fish and a robotic vehicle.

The manuscript proposes a technology that is potentially generalizable to soft robotic systems, but more data need to be presented. It is an exciting project, but missing much of the information that would qualify it for publication. Therefore, I would recommend acceptance given the following revisions:

Response:

Thank you very much for taking your valuable time to review our manuscript in your busy schedule. It is of great honor for us to get your professional and helpful comments to improve our manuscript. According to your constructive comments, we have made corresponding modifications and added new necessary data/results. Point-by-point corrections/responses are listed below.

Comment 1

The author claimed self-healing as a main feature of the pump and explained the working principles. There is, however, no evidence in the form of data or photo, that the self-healing function proposed is realized. The figures only show schematics of working principles. I suggest the author to present data that support the self-healing claim.

Answer 1

Thank you for your valuable comment. According to your helpful suggestion, we have added corresponding data/results about self-healing function.

(1) We conducted the dynamic thermomechanical analysis (DMA) test of the formed solid film after self-healing at different times and temperatures to show its viscoelastic properties. The DMA test was carried out in the thermodynamic analyzer (RSA-G2, TA), and the thickness and width of the tested self-healed film were 1 mm and 10 mm, respectively. The duration, oscillation strain, and frequency of the DMA test was set as 60 s, 5 %, and 10 Hz, respectively. The DMA results are shown in Supplementary Fig. 5. It can be seen from figure that the self-healed film is transparent and viscous. The storage modulus of the self-healed film is ~ 14.3 kPa at 20 °C, ~ 13.2 kPa at 25 °C, and ~ 10.6 kPa at 30 °C, respectively. The loss modulus of the self-healed film is ~ -3.1 kPa at 20 °C, ~ -2.5 kPa at 25 °C, and ~ -2.1 kPa at 30 °C, respectively. It is apparent that the storage modulus and loss modulus decreased as the temperature increased. We have added these corresponding contents in revised manuscript (Marked in red on page 8, pages 23-24, and page 37).

In page 8, we have added

Viscoelastic properties of the self-healed films are measured by dynamic thermomechanical analysis (DMA) tests, as show in Supplementary Fig. 5. The storage modulus of the self-healed film is ~ 14.3 kPa at 20 °C, ~ 13.2 kPa at 25 °C, and ~ 10.6 kPa at 30 °C, respectively. The loss modulus of the self-healed film is ~ -3.1 kPa at 20 °C, ~ -2.5 kPa at 25 °C, and ~ -2.1 kPa at 30 °C, respectively.

In pages 23-24, we have added

Viscoelastic properties of the self-healed films were measured by dynamic thermomechanical analysis (DMA) tests using thermodynamic analyzer (RSA-G2, TA), as shown in Supplementary Fig. 5. Thickness and width of the self-healed film were 1 mm and 10 mm, respectively. The duration, oscillation strain, and frequency of the DMA test was set as 60 s, 5 %, and 10 Hz, respectively. To evaluate the reliability, accuracy, and repeatability of experimental results, we tested three samples and each sample was tested three times. The DMA results are shown in Supplementary Fig. 5. It can be seen from figure that the self-healed film is transparent and viscous. The storage

modulus of the self-healed film is ~ 14.3 kPa at 20°C , ~ 13.2 kPa at 25°C , and ~ 10.6 kPa at 30°C , respectively. The loss modulus of the self-healed film is ~ -3.1 kPa at 20°C , ~ -2.5 kPa at 25°C , and ~ -2.1 kPa at 30°C , respectively. It is apparent that the storage modulus and loss modulus decreased as the temperature increased.

In page 37, we have added

Supplementary Fig. 5 | DMA tests at different time and temperature of self-healed films. a, Storage modulus of self-healed films at 20°C , 25°C , and 30°C . **b,** Loss modulus of self-healed films at 20°C , 25°C , and 30°C .

(2) To further characterize the adhesion property between the self-healed film and the silicone film, we conducted the repeated tensile tests of a silicone film with local self-healed film (shown in Supplementary Fig. 6). In the tests, the silicon film was repeatedly stretched for 200 times with 20% strain per time. After tests, it was found that the silicone film and the self-healed film were still firmly bonded together, illustrating good adhesion property between two films and the mechanical property of the self-healing performance. We have added these corresponding contents in revised manuscript (Marked in red on page 8, page 24, and page 38).

In page 8, we have added

To further characterize the adhesion property between the self-healed film and the silicone film, we conduct the repeated tensile tests of a silicone film with local self-

healed film, as show in Supplementary Fig. 6. The silicon film is repeatedly stretched for 200 times with 20% strain per time. After tests, it is found that the silicone film and the self-healed film are still firmly bonded together, illustrating good adhesion property between two films and the mechanical property of the self-healing performance.

In page 24, we have added

To further characterize the adhesion property between the self-healed film and the silicone film, we conducted the repeated tensile tests of a silicone film with local self-healed film, as shown in Supplementary Fig. 6. The self-healing liquid was dropped into a 1-mm-thickness silicone sheet, and then solidified to form a self-healed film bonding with the silicone sheet. The silicone sheet bonding with a self-healed film was cut into a size of $\sim 30 \text{ mm} \times 10 \text{ mm} \times 1 \text{ mm}$, and then was used to conduct tensile test (Universal tensile testing machine, ZQ-990B, Dongguan Zhiqu precision instruments Co., Ltd). In the tests, the silicon film was repeatedly stretched for 200 times with 20% strain per time. After tests, it was found that the silicone film and the self-healed film were still firmly bonded together, illustrating good adhesion property between two films and the mechanical property of the self-healing performance.

In page 38, we have added

Supplementary Fig. 6 | Tensile test of a silicone sheet bonding with a self-healed film.

(3) A self-healing process of is shown in Supplementary Video 3, where the self-healing function is realized by the formation of a cured film. Thus, the original leaking (damaged) region of the fluid power system is repaired by the self-healed film. Similarly, we have demonstrated the self-healing process in the robotic vehicle, as shown in Supplementary Fig. 12c. After self-healing, the robotic vehicle can continue to work without liquid leakage (Marked in red on page 17, page 31, and page 45).

In page 17, we have added

Supplementary Fig. 12c shows the self-healing process in the robotic vehicle.

In page 31, we have added

Supplementary Fig. 12c shows the self-healing process in the robotic vehicle.

In page 45, we have added

Supplementary Fig. 12 | Robotic vehicle. c, Self-healing process of the robotic vehicle.

Comment 2

One major issue is the writing. While this work presents some good results such as the

pump performance, the writing is structured in an unorganized way that is hard to follow.

Comment 2-1

a. The abstract and introduction does not highlight the contribution of this paper. The author tries to present the pump to be advantageous in every way (“fully soft, powerful, controllable, rapid response, built-in, long-term, and self-healing”) but the reader is lost in the real highlight of the contribution.

Answer 2-1

Thank you for your valuable comment. We have revised the abstract and introduction sections to highlight the contributions of our work.

Two main points are highlighted:

(1) Our soft pumps combine good portability and excellent actuation performance into one.

(2) The special functional liquids that merge unique properties of electrically actuation and self-healing function are developed, providing a direction for future self-healing fluid power systems.

We have revised the manuscript (Marked in red on page1 and page 3).

In page 1, we have revised

The soft pumps combine good portability with excellent actuation performance into one. We also develop the special functional liquids that merge unique properties of electrically actuation and self-healing function, providing a direction for future self-healing fluid power systems. Appearances and pumpabilities of soft pumps could be customized to meet personalized needs of diverse robots.

In page 3, we have revised

These soft pumps show some distinctive features: (i) They are not only lightweight and portable, but also exhibit powerful and controllable actuation capabilities; (ii) their appearances be easily customized to desired ones and implanted into different untethered soft robotic systems; (iii) their self-healing liquid can repair the damaged regions of fluidic systems, greatly improving the reliability of systems.

Comment 2-2

b. In the introduction, the author is too generalized in the background introduction by saying every mechanism has their unique disadvantage. A good literature review such as later discussed in line 222-241 should be summarized at the introduction to give readers an idea of the problem this paper is addressing.

Answer 2-2

Thank you for your valuable comment. We have summarized the advantages and disadvantages of each technique in the revised introduction (Marked in red on page 2).

In page 2, we have revised

To actuate soft robots, various soft actuation techniques⁵⁻⁷, including pressure⁸⁻¹⁰, thermal^{11,12}, magnetism^{13,14}, light¹⁵, combustion^{16,17}, phase transition¹⁸, etc., have been exploited. Traditional pneumatic/hydraulic soft actuators⁹ are most prevalent, yet the requirement of external bulky compressors/pumps¹⁹ leads to their poor portability, prohibiting their untethered applications. Recent advance in the stretchable electrohydrodynamic pumps²⁰ provide a way to solve the poor portability problem of traditional soft fluidic actuators, but their limited flow rate and output force cannot meet the actuation requirements of most untethered soft robots. Soft combustion-driven pumps¹⁷ can provide high-speed response and large generated pressure, but these are difficult to control and reuse. Thermal-activated phase transition¹⁸ driving power sources have the advantages of large output force, but their response speeds are slow due to poor controllability of thermal. Magnetic- or light-responsive¹³⁻¹⁵ soft actuation has unique advantages in driving micro/nano-scale robots, but the complex or bulky external equipment is always necessary.

Comment 2-3

c. The bioinspiration from spiders could make sense if the author is more clear about what is the similarity. Is the needle-hole structure inspired by the spider's heart? Or is the pump mimicking the functions of pumping and self-healing?

Answer 2-3

Thank you for your valuable comment. Our soft pumps mainly mimic the functions of pumping and self-healing, and we have added this point in the revised manuscript (Marked in red on pages 2-3).

In pages 2-3, we have added/revised

Is it possible for soft robots to possess driving power sources mimicking the functions of spiders' hearts that are fully soft, powerful, rapid response, built-in, diverse, long-term, quiet and even self-healing, driving their untethered motions and healing their damages?

To mimic the pumping and self-healing functions of spiders' hearts, here we introduce a class of fully soft electronic pumps that take advantage of electron and ion migration mechanism to pump liquid under applied electric field and are capable of automatically healing the damages of soft robots with help of self-healing liquid.

Comment 2-4

d. There needs to be a design section after the working principle is introduced to explain better each parameter's effect on the pumping performance of a single module. The author has this study but it's buried later in the text. I suggest designating a paragraph that discuss the findings in Supplementary Fig. 7.

Answer 2-4

Thank you for your valuable comment. According to your suggestion, we had added a paragraph to discuss the influence of each parameter on the pumping performance of a single module in the Main Text. (Marked in red on page 12).

In page 12, we have added

Parameters of electrode configurations (including needle diameter, hole diameter, and electrode gap) have obvious effects on the pumping performance (generated pressure and flow rate). Supplementary Figs. 9a-c show the influence of the needle

diameter, hole diameter, and electrode gap on the generated pressure. It is apparent that the generated pressure increases with the decreasing of needle diameter (Supplementary Fig. 9a), hole diameter (Supplementary Fig. 9b), and electrode gap (Supplementary Fig. 9c), respectively. Supplementary Figs. 9d-e show the influence of the needle diameter and electrode gap on the flow rate. The flow rate increases with the decreasing of needle diameter (Supplementary Fig. 9d) and electrode gap (Supplementary Fig. 9e), respectively.

Comment 2-5

e. There are lots of redundancy throughout the writing. Such as line 30-30.

Answer 2-5

Thank you for your valuable comment. We have tried our best to delete the redundancy throughout the manuscript.

Comment 3

Fig. 2 are all schematics. I suggest bringing the photos of design variations to Fig. 2.

Answer 3

Thank you for your comment. To make the designs of pump structures clear, in main text we present the schematics in Fig.2. According to these designs in Fig. 2, we had also fabricated these corresponding pumps. The photos of these real pumps are shown in Supplementary Fig. 7 in the manuscript. Considering the limited number of pictures in the journal and the color difference between these two Figures, we did this layout. If the figures are needed to further adjustment, we are very happy to continue to modify them.

In page 39

Supplementary Fig. 7 | Photographs of different soft pumps. The shells of the pumps are fabricated by 3D printing using thermoplastic elastomer (TPE) with a shore hardness 80 A.

Comment 4

Fig. 3b has a mismatching color for pump 1.

Answer 4

Thank you for your kind reminding, we have revised the color in Fig. 3b (Page 14).

Comment 5

Are there self-healing demonstrations in the robotic fish and vehicle? The conclusion seems to suggest that but I haven't found it in the robotic demonstrations.

Answer 5

Thank you for your kind comment. We had added an example of the self-healing results of robotic vehicle in Supplementary Fig. 12c. Soft actuator, which are core components of robotic fish and vehicle, can be self-healed in air after damage because a solid self-healed film will be formed when the self-healing functional liquids are in contact with air. We also conducted the self-healing test of robotic fish in water, but the experimental results showed that in water the robotic fish could not self-heal due to the lack of air. We have added the detailed discussion of self-healing in the revised manuscript (Marked in red on page 8, page 17, page 31, and page 45).

In page 8, we have added

The self-healing process can be achieved in air, but cannot be achieved in water due to lack of contact with air in water.

In page 17, we have added

Supplementary Fig. 12c shows the self-healing process in the robotic vehicle.

In page 31, we have added

Supplementary Fig. 12c shows the self-healing process in the robotic vehicle.

In page 45, we have added

Supplementary Fig. 12 | Robotic vehicle. c, Self-healing processes of the robotic vehicle.

Comment 6

More detailed improvements are needed here:

Comment 6-1

- Page 5, figure 1e and page 7, line 107: In figure 1e, the authors demonstrated the jet flow caused by electrical field, and only showed positive ions in the liquid. The authors further explained in page 7 that the mechanism of ion generation is through “positively charged process”. However, for different kinds of functional fluids, the dominant ion species could be either positive or negative. For example, in figure 1b from reference [15], negatively charged ions were the dominant species (Cacucciolo, V. et al. Stretchable pumps for soft machines. *Nature* 572, 516–519 (2019).). Ref [25] reported a way to distinguish dominant species via Sumoto effect (Kurobosh, Y., Takemura, K. & Edamura, K. Understanding of electro-conjugate fluid flow with dibutyl decanedioate using numerical simulation—Calculating ion mobility using molecular dynamics simulation. *Sens. Actuators B Chem.* 255, 448–453 (2018).). The reviewer would like to know if figure 1e is valid for both functional fluids (i.e., dibutyl sebacate and linalyl acetate) used in this paper ?

Answer 6-1

Thank you for your kind comment. So far, the dominant ion species are mainly distinguished by the flow direction of liquids in electric fields because the fluid flow is caused by the dragging of dominant ion species. If the flow direction is from positive electrode to negative electrode, dominant ion specie is positive; on the contrary, if the flow direction is from negative electrode to positive electrode, dominant ion specie is negative. In figure 1b from reference [15], negatively charged ions were the dominant species (Cacucciolo, V. et al. Stretchable pumps for soft machines. *Nature* 572, 516–519 (2019).), because their flow direction is from negative electrode to positive electrode. In our soft electronic pump, dominant ion species are positive ions because the flow direction is always from positive electrode to negative electrode, as shown in **Supplementary Video 2**.

The dominant species depend on functional fluids and electrode configurations. In our electrode configurations and two kind functional fluids, the dominant species are positive ions. To verify the operational principle, we propose theoretical model and numerical simulation in Methods and Supplementary Fig. 4, and the results show that

the theoretical model and numerical simulation of soft electronic pump are reliable and correct, so figure 1e is valid for both functional fluids (i.e., dibutyl sebacate and linalyl acetate) used in this paper.

Comment 6-2

- Page 8, line 152: the authors claimed that it took ~ 6 hours to achieve complete self-healing process under ambient temperature and the healed system could work for another 2 hours without leakage. Since self-healing is achieved by solidification of tung oil, which has a different composition from silicone, the reviewer would like to see a more quantitative way to characterize mechanical property of the pump.

Answer 6-2

Thank you for your valuable comment. Due to the small amount of liquid lost in the self-healing process, there is no obvious change in the pumping ability of the pump before and after self-healing. Self-healing process mainly affect the silicone film of fluid power systems. A self-healing film was formed to repair the damaged silicone film, so that the fluid power systems driven by the soft pumps can continue to work. Thus, we prepared a silicone film with local self-healed film in its center. To characterize the adhesion property between the self-healed film and the silicone film, we conducted the repeated tensile tests of a silicone film with local self-healed film (shown in Fig. 2). In the tests, the silicon film was repeatedly stretched for 200 times with 20% strain per time. After tests, it was found that the silicone film and the self-healed film were still firmly bonded together, illustrating good adhesion property between two films and the mechanical property of the self-healing performance. We have added these corresponding contents in revised manuscript (Marked in red on page 8, page 24, and page 38).

In page 8, we have added

To further characterize the adhesion property between the self-healed film and the silicone film, we conduct the repeated tensile tests of a silicone film with local self-healed film, as show in Supplementary Fig. 6. The silicon film is repeatedly stretched

for 200 times with 20% strain per time. After tests, it is found that the silicone film and the self-healed film are still firmly bonded together, illustrating good adhesion property between two films and the mechanical property of the self-healing performance.

In page 24, we have added

To further characterize the adhesion property between the self-healed film and the silicone film, we conducted the repeated tensile tests of a silicone film with local self-healed film, as shown in Supplementary Fig. 6. The self-healing liquid was dropped into a 1-mm-thickness silicone sheet, and then solidified to form a self-healed film bonding with the silicone sheet. The silicone sheet bonding with a self-healed film was cut into a size of $\sim 30 \text{ mm} \times 10 \text{ mm} \times 1 \text{ mm}$, and then was used to conduct tensile test (Universal tensile testing machine, ZQ-990B, Dongguan Zhiqu precision instruments Co., Ltd). In the tests, the silicon film was repeatedly stretched for 200 times with 20% strain per time. After tests, it was found that the silicone film and the self-healed film were still firmly bonded together, illustrating good adhesion property between two films and the mechanical property of the self-healing performance.

In page 38, we have added

Supplementary Fig. 6 | Tensile test of a silicone sheet bonding with a self-healed film.

Comment 6-3

- Page 10, line 180: supplementary fig. 6 is not called out throughout the main text. It should be placed somewhere in this paragraph.

Answer 6-3

According to your valuable suggestion, we have added supplementary fig. 6 (now is supplementary fig. 8) in Main text (Marked in red on pages 11).

In page 11, we have added

Supplementary Fig. 8 shows experimental setups for testing the performances of the soft electronic pump.

Comment 6-4

- Page 11, line 223 and page 13 figure 3a: the authors mentioned that the soft electronic pumps could reach similar pressure (~ 60 kPa) to current soft combustion-

driven pumps. However, there isn't enough experimental evidence indicating that the soft pump can reach ~ 60 kPa. In figure 1a, the highest pressure on the plot is below 10 kPa. Furthermore, the data of Pump 3 and Pump 4 under 14 kV and 16 kV are missing. The reviewer would like to know the reason.

Answer 6-4

Thank you for your valuable comment. We are sorry for this inappropriate expression. We have revised this sentence and added the missing information. Series integration of multiple needle-hole electrode pairs could obviously raise the generated pressure (Pages 11, lines 211-212). Here we mean that the soft electronic pump integrating nine needle-hole electrode pairs in series could achieve the same pressure (~ 60 kPa) (Marked in red on pages 13).

The reason why the data of Pump 3 and Pump 4 under 14 kV and 16 kV are missing is that the electrode gap between needle and hole electrodes of pump 3 and pump 4 is 0.8 mm, which is less than 2 mm in pump 1 and pump 2. The smaller gap between the electrodes results in dielectric breakdown at 14 kV for pump 3 and pump 4, while the dielectric breakdown did not happen in pump 1 and pump 2 with larger electrode gap. We have added this information in the revised manuscript (Marked in red on pages 15).

In page 13, we have added

the soft electronic pump integrating nine needle-hole electrode pairs in series could achieve the same pressure (~ 60 kPa)

In page 15, we have added

The dielectric breakdown voltages of pump 1 and pump 2 are above 16 kV, while those of pump 3 and pump 4 are ~ 14 kV.

Comment 6-5

- Page 13, figure 3b: the line color of Pump 1 is inconsistent with the one used in figure 3a.

Answer 6-5

Thanks for your careful checks. We are sorry for our carelessness. We have revised this error in Fig. 3b (Page 14).

Comment 6-6

- Page 22, line 451: Please use a gradient operator.

Answer 6-6

Thank you for your helpful suggestion. We have used the gradient operator in Equations (Page 25-E6).

Comment 6-7

- Page 24, line 473: Please specify the data source of these physical properties.

Answer 6-7

According to your suggestion, we have added the detailed information of data source in the revised manuscript (Marked in red on pages 27).

In page 27, we have added

The density, viscosity, relative permittivity, and conductivity of dibutyl sebacate based functional liquid were tested based on our experimental apparatus. The density, viscosity, relative permittivity, and conductivity of dibutyl sebacate based functional liquid were $\sim 938 \text{ kg/m}^3$ (WLD-300A, Electronic density tester, Wanliduo), $7.5 \times 10^{-3} \text{ Pa s}$ (NDJ-5S-5T, Digital rotary viscometer, Shanghai Fangrui Instrument Co.Ltd.), ~ 4.5 (Vector network analyzer, coaxial reflection method, measurement frequency is 1GHz) $\sim 3.4 \times 10^{-10} \text{ S/m}$ (Liquid Conductivity Meter), respectively. The ionic mobility of dibutyl sebacate based functional liquid is obtained based on the method used in the paper²⁵ (where dibutyl decanedioate and dibutyl sebacate are the same liquids) and lots of result comparisons of experiments and simulations. The ionic mobility of dibutyl sebacate based functional liquid is $\sim 6.93 \times 10^{-8} \text{ m}^2/(\text{V s})$.

Comment 6-8

- Since one important design feature of this pump is that it can operate bidirectionally, as it was used in the robotic fish, the reviewer would like to see if the cyclability of this kind of pump. Moreover, the authors should also briefly discuss the frequency limits of this pump under AC signal.

Answer 6-8

Thank you for your valuable comment. The dynamic bidirectional pumpabilities (cyclability) of soft pumps under a 1 Hz switching square wave (AC signal) are showed in Supplementary Fig. 9f (Page 41). Supplementary Video 4 showed the cyclability of this kind of pump under the high voltage with frequency of 1/6 Hz (AC signal). The response time (peak time) of soft pump is ~ 0.45 s, and the switching response time of bidirectional pumping is ~ 0.58 s, as shown in Fig. 3e (Page 12, lines 220-222). We tested the frequency limits of this pump under AC signal, and found the frequency limits are ~ 10 Hz. We have added this information in the revised manuscript (Marked in red on pages 12).

In page 41

Supplementary Fig. 9 | Customizable pumpabilities of soft electronic pumps. f, Dynamic generated pressure of soft electronic pumps under the applied square wave of 16-kV amplitude and 1-Hz frequency.

In page 12, we have added

The frequency limits of this pump is tested as ~ 10 Hz.

Reviewer #2 (Remarks to the Author):

In summary: This manuscript (Customizing a self-healing, soft pump for robot) shows an electrohydrodynamic pump for soft robots. The pump contains a soft structure that contains a liquid dielectric layer and soft electrode. When a high voltage is applied between the electrodes, ions move from one electrode to another, and drag fluid to create a pumping effect. The pump is also able to self-heal from mechanical damage, as the liquid would slowly form a thin film when exposed to air. Furthermore, the authors showed applications as an untethered robotic fish and an untethered vehicle, and they stated the pump has promising qualities, such as fully soft architecture, powerful actuation, rapid response, long lifetime, good controllability, excellent portability, high robustness, versatility, self-healing, easy fabrication, and low-cost. Under my point of view, the data and results shown in this paper could further be improved, and there are some comments that the authors need to address. Thus, I believe this paper cannot be accepted in Nature Communications until more explanations are included in the corrected version of the submitted manuscript.

Response:

Thank you very much for taking your valuable time to review our manuscript in your busy schedule. We feel great thanks for your professional and valuable comments to improve our manuscript. According to your constructive comments, we have made corresponding modifications and added new experimental data/results. Point-by-point corrections/responses are listed below. Thanks again!

Comment 1

In the introduction, the authors referenced electrostatic force (ref. 15) have their own distant shortcomings when used in untethered soft systems. The authors should remove the electrostatic force, as this paper's mechanism is still based on electrostatic force mechanism, as the electric field causes positive ion migration.

Answer 1

Thank you for your helpful comment. We are really sorry for this inappropriate

expression. We have revised this content in the Introduction (Page 2).

In page 2, we have revised

To actuate soft robots, various soft actuation techniques⁵⁻⁷, including pressure⁸⁻¹⁰, thermal^{11,12}, magnetism^{13,14}, light¹⁵, combustion^{16,17}, phase transition¹⁸, etc., have been exploited.

Comment 2

To make the introduction more convincing, the authors should provide details on what they could do better than ref. 15 in the introduction. Many of these examples could be seen in Table 1, in which the authors provided for previous reviewer 1 for Nature Materials submission (such as Positive ions migration is higher than negative ions, much higher pressure of single electrode pair, less liquid dragging in the liquid flow).

Answer 2

We sincerely appreciate your valuable suggestion. According to your suggestion, we have added some sentences to summarize the contributions and limitations of Ref. 15's (now is Ref. 20) work in the Introduction section firstly, and then added our advantages over their work in the Main Text. Also we have added the corresponding Table 1 in the revised manuscript (Marked in red on page 2, pages 13-14, and page 47)

In page 2, we have added

Recent advance in the stretchable electrohydrodynamic pumps²⁰ provide a way to solve the poor portability problem of traditional soft fluidic actuators, but their limited flow rate and output force cannot meet the actuation requirements of most untethered soft robots.

In pages 13-14, we have added

Compared with stretchable electrohydrodynamic pumps²⁰, needle-hole electrode configuration of soft electronic pumps not only could generate much more powerful pressures (pressure of single electrode pair: soft electronic pump – 9.2 kPa > stretchable

pump – 7 kPa/34 = 0.21 kPa) and flow rates (soft electronic pump – 521 ml/min > stretchable pump – 100 μ l/s = 6 ml/min), but also benefits for flexible and diverse spatial arrangement, overcoming the shortcomings of insufficient output capacities, slow system responses (system response when the pumps are embedded into actuators: soft electronic pump – less than 1 s < stretchable pump – above 30 s), and monotonous appearances of stretchable pumps. **More detailed comparison of soft electronic pump and stretchable electrohydrodynamic pump are indicated in Supplementary Table 1.**

In page 47, we have added

	Soft electronic pump	Stretchable EHD pump²⁰
Underlying mechanism	Positive ions migration	Negative ions migration
Electrode structure	Needle-hole electrode pair (spatial structure)	Plane electrode pair (plane structure)
Electric field	Strong non-uniform electric field	Electric field in electrode pair is symmetric
Migration ion species	Positive ions	Negative ions
Speed of ion migration	Positive ions > Negative ions	
Flow form	Strong jet	EHD flow
Flow channel	Large cross-areas	Long and narrow channels
Flow direction	Positive electrode to grounding electrode	Grounding electrode to positive electrode
Number of electrode pairs in current version	4	34
Pressure of single electrode pair	~ 9.2 kPa	~ 7 kPa/34 = 0.21 kPa
Flow rate	~ 521 ml/min	~ 100 μ l/s/ = 6 ml/min
System response speed when the pumps are embedded into actuators	Less than 1 s	Above 30 s
Electrode materials	Conductive silicone materials long lifetime + good stretchability	C or Ag electrodes C – short lifetime (only 15 min) Ag – poor stretchability
Appearance	Arbitrary	Monotonous plane structure

Supplementary Table 1 | Comparison of soft electronic pump and stretchable electrohydrodynamic (EHD) pump²⁰

Comment 3

In the second paragraph of introductions, more detail is needed on how the haemolymph

works, such as explaining how the haemolymph could generate hydraulic pressure. I suggest the authors to provide more detail and also to cite Figure 1a in the introduction section.

Answer 3

According to your constructive suggestion, we have added more details about spiders' haemolymph in the revised manuscript and cited Figure 1a in the introduction section (Marked in red on page 2).

In page 2, we have added

The hydraulic power of haemolymph flow is generated by the hearts' pumping function, and then the pressurized hemolymph flows in their foot to achieve their movements, as shown in Fig. 1a.

Comment 4

In line 120-122, the authors mentioned “Such free electrons will be absorbed into needle positive electrodes from liquid, resulting in formation of free ions with positive charge.” It is difficult to visualize how these electrons absorbing into positive electrodes would form positive charged free ions. Please provide more details on how this positive charged free ions are formed.

Answer 4

Thank you for your valuable comment. We are very sorry for this inappropriate expression. The formation process of positive charged free ions is a kind of corona discharges in dielectric fluids. This sentence means that the strong electric field causes the electrons in a small amount of neutral liquid molecules near needle positive electrodes to overcome potential barriers and separate from neutral liquid molecules to become free electrons. These free electrons with negative charges will be absorbed into needle positive electrodes from liquid. The original neutral liquid molecules that lost electrons turned into positively charged ions (Marked in red on page 6). This principle is similar to that of ion propulsion aeroplane (Flight of an aeroplane with solid-state propulsion, Nature 563, 532-535 (2018)). The detailed principle is described as

“Electroaerodynamics (EAD) is a means of generating propulsive forces in fluids. Ions generated in the ambient fluid and under the influence of an applied electric field are accelerated by the Coulomb force. These ions collide with neutral molecules and couple the momentum of the accelerated ions with that of the bulk fluid; the result is an ionic wind that produces a thrust force in the opposite direction to ion flow. In our device, we generate ions using a corona discharge. A corona discharge is a self-sustaining atmospheric discharge that is induced by the application of a constant high electric potential across two asymmetric electrodes; high electric fields near the smaller electrode accelerate electrons and produce a cascade of ionization by successive electron collisions with neutral molecules.”

In page 6, we have revised

These free electrons with negative charges will be absorbed into needle positive electrodes from liquid. The original neutral liquid molecules become positive ions due to the loss of electrons, so-called positively charged process.

Comment 5

For self-healing, the authors states that it takes 6 hours at an ambient temperature (35 degrees C) for the self-healing process. However, 35 degrees C is not ambient temperature. I suggest the authors to perform the test at room temperature.

Answer 5

Thank you for your kind comment. We had performed the self-healing tests in two temperatures: ~ 6 hours at 35 °C and ~ 1 day at 24 °C (Previous version in the Method, page 23, line 459). Both of these two temperatures are room temperatures, but we did these two tests at different seasons. The temperature has an obvious effect on the self-healing time, and we have added this information in the Main Text (Marked in red on page 8).

In page 8, we have added

The temperature has an obvious effect on the self-healing time, and the times of

complete self-healing process of the punctured damage are ~ 6 hours at 35 °C and ~ 1 day at 24 °C, respectively.

Comment 6

Also, since the self-healing of the tung oil is a novelty of this work. I suggest the authors to do more characterization of the self-healed film, such as DMA tests at different time and temperature of self-healing.

Answer 6

According to your valuable suggestion, we also conducted the dynamic thermomechanical analysis (DMA) test of the formed solid film after self-healing at different times and temperatures to show its viscoelastic properties. The DMA test was carried out in the thermodynamic analyzer (RSA-G2, TA), and the thickness and width of the tested self-healed film were 1 mm and 10 mm, respectively. The duration, oscillation strain, and frequency of the DMA test was set as 60 s, 5 %, and 10 Hz, respectively. The DMA results are shown in the following Fig. 1. It can be seen from figure that the self-healed film is transparent and viscous. The storage modulus of the self-healed film is ~ 14.3 kPa at 20 °C, ~ 13.2 kPa at 25 °C, and ~ 10.6 kPa at 30 °C, respectively. The loss modulus of the self-healed film is ~ -3.1 kPa at 20 °C, ~ -2.5 kPa at 25 °C, and ~ -2.1 kPa at 30 °C, respectively. It is apparent that the storage modulus and loss modulus decreased as the temperature increased. We have added these corresponding contents in revised manuscript (Marked in red on page 8, pages 23-24, and page 37).

In page 8, we have added

Viscoelastic properties of the self-healed films are measured by dynamic thermomechanical analysis (DMA) tests, as show in Supplementary Fig. 5. The storage modulus of the self-healed film is ~ 14.3 kPa at 20 °C, ~ 13.2 kPa at 25 °C, and ~ 10.6 kPa at 30 °C, respectively. The loss modulus of the self-healed film is ~ -3.1 kPa at 20 °C, ~ -2.5 kPa at 25 °C, and ~ -2.1 kPa at 30 °C, respectively.

In pages 23-24, we have added

Viscoelastic properties of the self-healed films were measured by dynamic thermomechanical analysis (DMA) tests using thermodynamic analyzer (RSA-G2, TA), as shown in Supplementary Fig. 5. Thickness and width of the self-healed film were 1 mm and 10 mm, respectively. The duration, oscillation strain, and frequency of the DMA test was set as 60 s, 5 %, and 10 Hz, respectively. To evaluate the reliability, accuracy, and repeatability of experimental results, we tested three samples and each sample was tested three times. The DMA results are shown in Supplementary Fig. 5. It can be seen from figure that the self-healed film is transparent and viscous. The storage modulus of the self-healed film is ~ 14.3 kPa at 20 °C, ~ 13.2 kPa at 25 °C, and ~ 10.6 kPa at 30 °C, respectively. The loss modulus of the self-healed film is ~ -3.1 kPa at 20 °C, ~ -2.5 kPa at 25 °C, and ~ -2.1 kPa at 30 °C, respectively. It is apparent that the storage modulus and loss modulus decreased as the temperature increased.

In page 37, we have added

Supplementary Fig. 5 | DMA tests at different time and temperature of self-healed films. a, Storage modulus of self-healed films at 20 °C, 25 °C, and 30 °C. **b,** Loss modulus of self-healed films at 20 °C, 25 °C, and 30 °C.

Comment 7

The authors showed Figure 3a, but did not mention what are the four pumps being tested in the manuscript. The author should clarify by adding a table in the manuscript showing the difference of each pump that is used.

Answer 7

According to your valuable suggestion, we have added a table to clarify the difference of each pump in the revised manuscript (Marked in red on page 48).

In page 48, we have added

	Electrode configuration	Functional liquid
Pump 1	Electrode design 1	Liquid 1
Pump 2	Electrode design 1	Liquid 2
Pump 3	Electrode design 2	Liquid 1
Pump 4	Electrode design 2	Liquid 2
Electrode design 1 - 1-mm diameter needle, 3-mm diameter hole, and 2-mm gap between them		
Electrode design 2 - 0.4-mm diameter needle, 1-mm diameter hole, and 0.8-mm gap between them		
Liquid 1 - Dibutyl sebacate based functional liquid		
Liquid 2 - Linalyl acetate based functional liquid		

Supplementary Table 2 | Difference of four pumps used in Fig. 3. These four pumps are a combination of two kinds of electrode designs and two kinds of liquids.

Comment 8

The authors showed in Figure 3f showing that applying a voltage of 15kV for 4 hours, showing the durability and reliability of soft pumps. It could be seen that the pressure starts to decrease when holding the constant voltage to longer times. I suggest the authors should provide an explanation for why the pressure would decrease after applying steady voltage.

Answer 8

Thank you for your kind comment. The soft electrodes made of soft materials (silicon) would be passivated at the beginning of use, and then tend to be stabilized, thereby causing the pressure decreasing first and then stabilizing. A similar phenomenon could also be seen in Fig. 2c of Cacucciolo et al.'s work (Stretchable pumps for soft machines.

Nature 572, 516–519 (2019).).

We have added the detailed explanation in the revised manuscript (Marked in red on page 12).

In page 12, we have added

It is worth noting that the electrodes would be passivated and then stabilized when high voltage was applied in a longer times, thereby causing the pressure decreasing and then stabilizing.

Comment 9

In Figure 3g, the authors showed the power consumption of their work to be around 3-4 W. The authors should provide calculations on how they were able to obtain this value.

Answer 9

According to your valuable comment, we have added the detailed calculations of the power consumption value. The power consumption is calculated based on the measured electric current and voltage ($P = U \times I$). The maximum power consumption of the soft electronic pump is ~ 3.6 W (= Voltage ~ 16 kV \times Electric current ~ 225 μ A).

We have added this information in the revised manuscript (Marked in red on page 13).

In page 13, we have added

The maximum power consumption of the soft electronic pump is ~ 3.6 W (Voltage ~ 16 kV, electric current ~ 225 μ A).

Reviewer #1 (Remarks to the Author):

The technical improvements are satisfactory, except I will point out that the rebuttal including misspellings of "silicone" as "silicon," and the use of "viscous" when they meant "elastic." Please make sure these errors are not in the main text, I may have missed it.

I still do not understand the motivation behind a "spider's heart." Aren't all hearts self healing? Please remove this aspect of the paper or do better at convincing me that it actually motivated the work, instead of simply trying to make it sound more exciting. Right now it devalues the contribution.

Reviewer #2 (Remarks to the Author):

Summary

This manuscript (Customizing a self-healing, soft pump for robot) shows an electrohydrodynamic pump for soft robots. The pump contains a soft structure that contains a liquid dielectric layer and soft electrode. When a high voltage is applied between the electrodes, ions move from one electrode to another, and drag fluid to create a pumping effect. The pump is also able to self-heal from mechanical damage, as the liquid would slowly form a thin film when exposed to air. Furthermore, the authors showed applications as an untethered robotic fish and an untethered vehicle, and they stated the pump has promising qualities, such as fully soft architecture, powerful actuation, rapid response, long lifetime, good controllability, excellent portability, high robustness, versatility, self-healing, easy fabrication, and low-cost. In the rebuttal that the authors submitted, the authors provided point to point responses to my questions/comments. However, there are still minor comments that the authors need to address. After addressing the comments, I would recommend acceptance.

Comments to the Authors

1. In the introduction, the authors added 'Recent advance in the stretchable electrohydrodynamic pumps provide a way to solve the poor portability problem of traditional soft fluid actuators, but their limited flow rate and output force cannot meet the actuation requirements of most untethered robots'. However, I think this sentence does not fit into paragraph 2, as electrohydrodynamic force is not mentioned in paragraph 1 of the introduction. A better location for a similar sentence talking about electrohydrodynamic pumps would be at the start of paragraph 3, with the logic of what previous pumps has been done that can mimic spider's heart, their shortcomings, and then proposing your soft electronic pump that take advantage of both ion and electron migration for more powerful actuation, ability to be easily customized, and self-healing capabilities. With this edit, it could make the readers easily understand the highlights of the paper.
2. For the self-healing process, it would be good for the reader to know more about how the mechanical properties would change, or if it would change from the pristine to the self-healed samples. The authors should provide a comparison of pristine and self-healed samples, such as by possibly looking at the tensile stress-strain curve.
3. The experiments to run the self-healing tests should also be added in the methods section.

Detailed Responses to Reviewers

Reviewer #1 (Remarks to the Author):

Comment 1

The technical improvements are satisfactory, except I will point out that the rebuttal including misspellings of "silicone" as "silicon," and the use of "viscous" when they meant "elastic." Please make sure these errors are not in the main text, I may have missed it.

Answer 1

First of all, we would like to sincerely thank you for your valuable and constructive suggestions. These valuable suggestions improve the quality of our work significantly. Thank you for your kind reminder, we have carefully checked and revised the misspellings all through the manuscript. Especially, the “silicon” has been revised as “silicone” (Marked in red on page 8 and page 21); the “viscous” has been revised as “elastic” (Marked in red on page 8 and page 21).

Comment 2

I still do not understand the motivation behind a "spider's heart." Aren't all hearts self healing? Please remove this aspect of the paper or do better at convincing me that it actually motivated the work, instead of simply trying to make it sound more exciting. Right now it devalues the contribution.

Answer 2

Thank you for your valuable comment. We agree with you very much that all hearts are self-healing. In terms of self-healing function, spider's heart and other animals' hearts are the same. But compared with other animal hearts, spider's heart has its own special feature. The hearts of other animals (such as dog, cat) are mainly used to pump blood to transport nutrients, and their limbs' movement is driven by their muscles, not directly driven by the pressurized blood from their hearts. But for spider, the movement of its limbs is directly achieved by the pressurized haemolymph from its hearts, which is similar to hydraulic system. The movements of our robots are also directly achieved by

the pressurized liquid generated by our self-healing soft pumps inside the robots. Thus, the movement actuation of spider are similar that of our robots. The motivation of "spider's heart" is based on both self-healing function and this direct hydraulic actuation function. We are sorry that, in our original version, we used too few sentences to descript this point, which may be not very clear to cause readers' misunderstanding. In the revised version, we have added more sentences to make it clearer (Marked in red on page 2).

In page 2, we have added:

In nature, spiders have their unique biological hydraulic systems^{20,21}, allowing them to achieve autonomous and agile motions by directly utilizing hydraulic power of haemolymph flow. The hydraulic power of haemolymph flow is generated by the hearts' pumping function, and then the pressurized hemolymph flows in their foot to achieve their movements, as shown in Fig. 1a. Whereas other animals (such as dog, cat, etc.) use their hearts to pump blood so as to transport nutrients, and their limbs' movement is driven by muscles, not directly driven by the pressurized blood from their hearts.

Reviewer #2 (Remarks to the Author):

In summary: This manuscript (Customizing a self-healing, soft pump for robot) shows an electrohydrodynamic pump for soft robots. The pump contains a soft structure that contains a liquid dielectric layer and soft electrode. When a high voltage is applied between the electrodes, ions move from one electrode to another, and drag fluid to create a pumping effect. The pump is also able to self-heal from mechanical damage, as the liquid would slowly form a thin film when exposed to air. Furthermore, the authors showed applications as an untethered robotic fish and an untethered vehicle, and they stated the pump has promising qualities, such as fully soft architecture, powerful actuation, rapid response, long lifetime, good controllability, excellent portability, high robustness, versatility, self-healing, easy fabrication, and low-cost. In the rebuttal that the authors submitted, the authors provided point to point responses to my questions/comments. However, there are still minor comments that the authors need to address. After addressing the comments, I would recommend acceptance.

Response:

First of all, we sincerely thank you for your constructive and positive comments to improve our manuscript. These valuable suggestions improve the quality of our work significantly. According to your constructive comments, we have made corresponding modifications and added new experimental data/results. Point-by-point corrections/responses are listed below. Thanks again!

Comment 1

In the introduction, the authors added ‘Recent advance in the stretchable electrohydrodynamic pumps provide a way to solve the poor portability problem of traditional soft fluid actuators, but their limited flow rate and output force cannot meet the actuation requirements of most untethered robots’. However, I think this sentence does not fit into paragraph 2, as electrohydrodynamic force is not mentioned in paragraph 1 of the introduction. A better location for a similar sentence talking about electrohydrodynamic pumps would be at the start of paragraph 3, with the logic of what

previous pumps has been done that can mimic spider's heart, their shortcomings, and then proposing your soft electronic pump that take advantage of both ion and electron migration for more powerful actuation, ability to be easily customized, and self-healing capabilities. With this edit, it could make the readers easily understand the highlights of the paper.

Answer 1

Thank you for your helpful comment. Indeed, your suggestion is more conducive to improving the presentation of the paper. According to your valuable suggestion, we have revised this content in the Introduction (Marked in red on page 3).

In page 3, we have revised:

Recent stretchable electrohydrodynamic pumps²³ that can be embedded in soft robots provide a good way to solve the poor portability problem of traditional soft fluidic actuators, which can mimic the pumping function of spiders' hearts. Nevertheless, their flow rate and output force should be further improved to meet the actuation requirements of most untethered soft robotic applications. To mimic both pumping and self-healing functions of spiders' hearts, here we introduce a class of fully soft electronic pumps that take advantage of electron and ion migration mechanism to pump liquid under applied electric field and are capable of automatically healing the damages of soft robots with help of self-healing liquid.

Comment 2

For the self-healing process, it would be good for the reader to know more about how the mechanical properties would change, or if it would change from the pristine to the self-healed samples. The authors should provide a comparison of pristine and self-healed samples, such as by possibly looking at the tensile stress-strain curve.

Answer 2

According to your valuable suggestion, we conducted the tensile stress-strain test of pristine and self-healed samples. The tensile stress-strain curves of pristine and self-healed samples are close, illustrating that the difference in the mechanical properties

between them is small (Marked in red on page 8, page 21 and Supplementary information page 7).

In page 8, we have added:

The tensile stress-strain curves of pristine and self-healed samples are close, as show in Supplementary Fig. 6b, illustrating that the difference in the mechanical properties between them is small.

In page 21, we have added:

To compare the mechanical properties of pristine and self-healed samples, uniaxial tensile tests were conducted at an ASTM D638 (Type IV) universal test machine with a crosshead speed of 50 mm/min. Supplementary Fig. 6b showed that the uniaxial tensile stress-strain curves of pristine and self-healed samples are close, illustrating that the difference in the mechanical properties between them is small. Note that the applied stresses of the self-healing sample are slightly greater than those of the pristine sample to achieve the same strain due to that the local self-healing film makes the self-healing sample slightly harder than the pristine sample.

In Supplementary information page 7, we have added:

Supplementary Fig. 6 | Tensile tests. a, Tensile test of a silicone sheet bonding with a self-healed film. **b**, Tensile stress-strain curves of pristine and self-healed samples.

Comment 3

The experiments to run the self-healing tests should also be added in the methods section.

Answer 3

According to your kind suggestion, we have added the experiments to run the self-healing tests in the methods section (Marked in red on page 20).

In page 20, we have added:

A set number of self-healing droplets were attached to silicone sheets, and a small glass rod was used to touch the self-healing area every hour to evaluate whether the self-healing liquid is solidified. At the same time, a timer was used to record the self-healing time.